# The theory of massively repeated evolution and full identifications of cancer-driving nucleotides (CDNs)

Lingjie Zhang[1], Tong Deng[1], Zhongqi Liufu[1,2], Xueyu Liu[1], Bingjie Chen[1,3], Zheng Hu[4], Chenli Liu[4], Miles E Tracy[1], Xuemei Lu[2]*, Hai-Jun Wen[1,5]*, Chung-I Wu[1,5,6]*

[1]State Key Laboratory of Biocontrol, School of Life Sciences, Sun Yat-sen University, Guangzhou, China; [2]State Key Laboratory of Genetic Resources and Evolution/ Yunnan Key Laboratory of Biodiversity Information, Kunming Institute of Zoology, The Chinese Academy of Sciences, Kunming, China; [3]GMU-GIBH Joint School of Life Sciences, Guangzhou Medical University, Guangzhou, China; [4]CAS Key Laboratory of Quantitative Engineering Biology, Shenzhen Institute of Synthetic Biology, Institute of Advanced Technology, Chinese Academy of Sciences, Shenzhen, China; [5]Innovation Center for Evolutionary Synthetic Biology, Sun Yat-sen University, Guangzhou, China; [6]Department of Ecology and Evolution, University of Chicago, Chicago, United States

*For correspondence:
xuemeilu@mail.kiz.ac.cn (XL);
wenhj5@mail.sysu.edu.cn (H-JW);
ciwu@uchicago.edu (C-IW)

Competing interest: The authors declare that no competing interests exist.

## eLife Assessment

This **important** paper introduces a theoretical framework and methodology for identifying Cancer Driving Nucleotides (CDNs), primarily based on single nucleotide variant (SNV) frequencies. A variety of **solid** approaches indicate that a mutation recurring three or more times is more likely to reflect selection rather than being the consequence of a mutation hotspot. The method is rigorously quantitative, though the requirement for larger datasets to fully identify all CDNs remains a noted limitation. The work will be of broad interest to cancer geneticists and evolutionary biologists.

**Abstract** Tumorigenesis, like most complex genetic traits, is driven by the joint actions of many mutations. At the nucleotide level, such mutations are cancer-driving nucleotides (CDNs). The full sets of CDNs are necessary, and perhaps even sufficient, for the understanding and treatment of each cancer patient. Currently, only a small fraction of CDNs is known as most mutations accrued in tumors are not drivers. We now develop the theory of CDNs on the basis that cancer evolution is massively repeated in millions of individuals. Hence, any advantageous mutation should recur frequently and, conversely, any mutation that does not is either a passenger or deleterious mutation. In the TCGA cancer database (sample size $n$=300–1000), point mutations may recur in $i$ out of $n$ patients. This study explores a wide range of mutation characteristics to determine the limit of recurrences ($i^*$) driven solely by neutral evolution. Since no neutral mutation can reach $i^*$=3, all mutations recurring at $i{\geq}3$ are CDNs. The theory shows the feasibility of identifying almost all CDNs if $n$ increases to 100,000 for each cancer type. At present, only <10% of CDNs have been identified. When the full sets of CDNs are identified, the evolutionary mechanism of tumorigenesis in each case can be known and, importantly, gene targeted therapy will be far more effective in treatment and robust against drug resistance.

## Introduction

Cancers are complex genetic traits with multiple mutations that interact to yield the ensemble of tumor phenotypes. The ensemble has been characterized as 'cancer hallmarks' that include sustaining growth signaling, evading growth suppression, resisting apoptosis, achieving immortality, executing metastasis and so on *Hanahan and Weinberg, 2000*; *Hanahan and Weinberg, 2011*; *Hanahan, 2022*. It seems likely that each of the 6–10 cancer hallmarks is governed by a set of mutations. Most, if not all, of these mutations are jointly needed to drive the tumorigenesis.

In the genetic sense, cancers do not differ fundamentally from other complex traits whereby multiple mutations are simultaneously needed to execute the program. A well-known example is the genetics of speciation whereby interspecific hybrids are either sterile or infertile even though they do not have deleterious genes (*Wu and Ting, 2004*; *Wang et al., 2022*; *Wu, 2022*). A recent example is SARS-CoV-2. The early onset of COVID-19 requires all four mutations of the D614G group and the later Delta strain has 31 mutations accrued in three batches (*Ruan et al., 2022b*; *Ruan et al., 2022a*; *Cao et al., 2023*; *Ruan et al., 2023*) While cancer research has often proceeded one mutation at a time, each of the mutations has been shown to be insufficient for tumorigenesis until many (*Ortmann et al., 2015*; *Takeda, 2021*; *Hodis et al., 2022*) are co-introduced.

We now aim for the identification of all (or at least most) of the driver mutations in each patient. Both functional tests and treatments demand such identifications. The number of key drivers has been variously estimated to be 6–10 (*Martincorena et al., 2017*; *Anandakrishnan et al., 2019*; *ICGC/TCGA Pan-Cancer Analysis of Whole Genomes Consortium, 2020*). Although cancer driving 'point mutations', referred to as Cancer Driving Nucleotides (or CDNs), are not the only drivers, they are indeed abundant (*ICGC/TCGA Pan-Cancer Analysis of Whole Genomes Consortium, 2020*). Furthermore, CDNs, being easily quantifiable, may be the only type of drivers that can be fully identified (see below). Here, we will focus on the clonal mutations present in all cells of the tumor without considering within-tumor heterogeneity for now (*Ling et al., 2015*; *Turajlic et al., 2019*; *Black and McGranahan, 2021*; *Chen et al., 2022a*; *Zhai et al., 2022*; *Bian et al., 2023*; *Zhu et al., 2023*).

Since somatic evolution proceeds in parallel in millions of humans, point mutations can recur multiple times as shown in *Figure 1*. The recurrences should permit the detection of advantageous mutations with unprecedented power. The converse should also be true that mutations that do not recur frequently are unlikely to be advantageous. *Figure 1* depicts organismal evolution and cancer

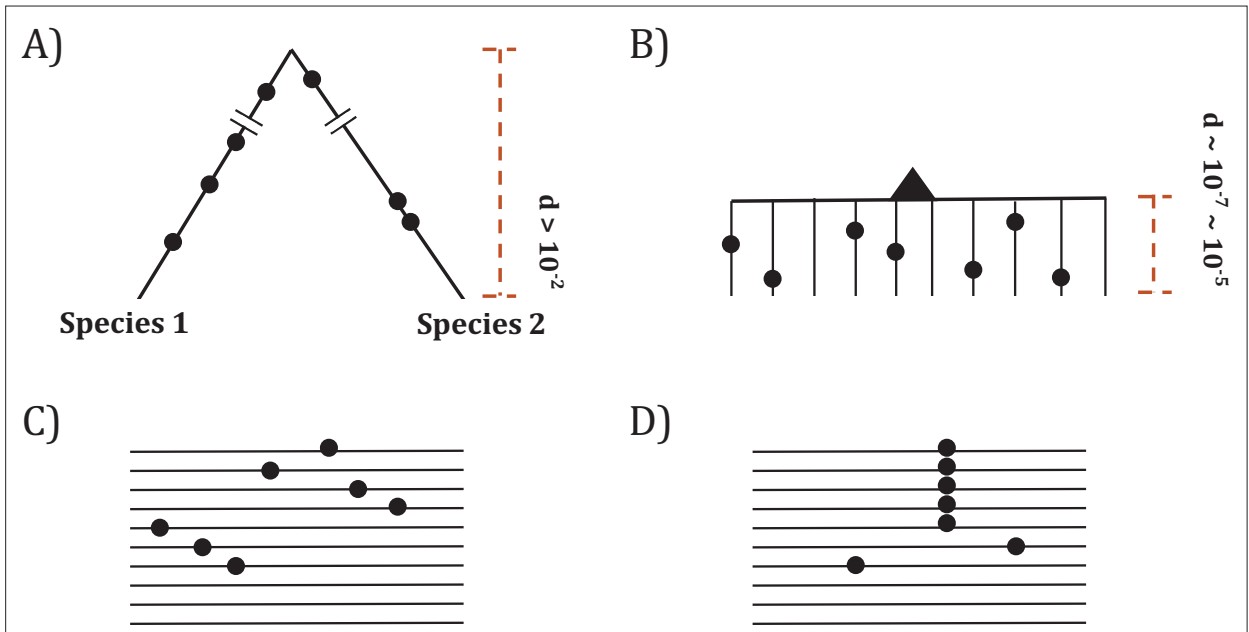

**Figure 1.** Two modes of DNA sequence evolution. (**A**) A hypothetical example of DNA sequences in organismal evolution. (**B**) Cancer evolution that experiences the same number of mutations as in (**A**) but with many short branches. (**C**) A common pattern of sequence variation in cancer evolution. (**D**) In cancer evolution, the same mutation at the same site may occasionally be seen in multiple sequences. The recurrent sites could be either mutational or functional hotspots, their distinction being the main objective of this study.

evolution. While both panels **A** and **B** show 7 mutations, there can be two patterns for cancer evolution - the pattern in (**C**) where all mutations are at different sites is similar to the results of organismal evolution whereas the pattern in (**D**) is unique in cancer evolution. The hotspot of recurrences shown in (**D**) holds the key to finding all CDNs in cancers.

In the literature, hotspots of recurrent mutations have been commonly reported (*Gartner et al., 2013*; *Chang et al., 2016*; *Cannataro et al., 2018*; *Buisson et al., 2019*; *Hess et al., 2019*; *Stobbe et al., 2019*; *Juul et al., 2021*; *Nesta et al., 2021*; *Zhao et al., 2021*; *Bergstrom et al., 2022*; *Sherman et al., 2022*; *Wong et al., 2022*; *Zeng and Bromberg, 2022*). A hotspot, however, could be either a mutational or functional hotspot. Mutational hotspots are the properties of the mutation machinery that would include nucleotide composition, local chromatin structure, timing of replication, etc. (*Stamatoyannopoulos et al., 2009*; *Pleasance et al., 2010*; *Makova and Hardison, 2015*; *Polak et al., 2015*; *Martincorena et al., 2017*). In contrast, functional hotspots are CDNs under positive selection. CDN evolution is akin to 'convergent evolution' that repeats itself in different taxa (*He et al., 2020a*; *He et al., 2020b*; *Wu et al., 2020*; *Pan et al., 2022b*; *Wu, 2023*) and is generally considered the most convincing proof of positive selection.

While many studies conclude that sites of mutation recurrence are largely mutational hotspots (*Buisson et al., 2019*; *Hess et al., 2019*; *Stobbe et al., 2019*; *Nesta et al., 2021*; *Bergstrom et al., 2022*), others deem them functional hotspots, driven by positive selection (*Gartner et al., 2013*; *Chang et al., 2016*; *Bailey et al., 2018*; *Cannataro et al., 2018*; *Juul et al., 2021*; *Zhao et al., 2021*; *Zeng and Bromberg, 2022*). In the attempt to distinguish between these two hypotheses, studies make assumptions, often implicitly, about the relative importance of the two mechanisms in their estimation procedures. The conclusions naturally manifest the assumptions made when extracting information on mutation and selection from the same data (*Elliott and Larsson, 2021*).

This study consists of three parts. First, the mutational characteristics of sequences surrounding CDNs are analyzed. Second, a rigorous probability model is developed to compute the recurrence level at any sample size. Above a threshold of recurrence, all mutations are CDNs. Third, we determine the necessary sample sizes that will yield most, if not all, CDNs. In the companion study, the current cancer genomic data are analyzed for the characteristics of CDNs that have already been discovered (*Zhang et al., 2024*). Together, these two studies show how full functional tests and precise target therapy can be done on each cancer patient.

**Table 1.** An example of $A_i$ and $S_i$ (from lung cancer, n=1035).

| $i$ | All sites | | | CpG sites removed | | |
| --- | --- | --- | --- | --- | --- | --- |
| | $A_i$ | $S_i$ | $A_i / S_i$ | $A_i$ | $S_i$ | $A_i / S_i$ |
| 0 | 22540623 | 7804281 | 2.89 | 21375384 | 7014012 | 3.04 |
| 1 | 195958 | 69393 | 2.82 | 168371 | 56821 | 2.96 |
| 2 | 2946 | 969 | 3.04 | 2188 | 643 | 3.4 |
| 3 | 99 | 21 | 4.71 | 68 | 16 | 4.25 |
| 4 | 23 | 1 | 23 | 17 | 1 | 17 |
| 5 | 16 | 0 | 16 : 0 | 9 | 0 | 9 : 0 |
| 6 | 10 | 0 | 10 : 0 | 6 | 0 | 6 : 0 |
| 7 | 5 | 0 | 5 : 0 | 5 | 0 | 5 : 0 |
| 8 | 8 | 0 | 8 : 0 | 6 | 0 | 6 : 0 |
| 9 | 4 | 0 | 4 : 0 | 3 | 0 | 3 : 0 |
| ≥3 | 178 | 22 | 8.09 | 122 | 17 | 7.18 |
| ≥4 | 79 | 1 | 79 | 54 | 1 | 54 |
| [10-20] | 7 | 1 | 7 | 4 | 0 | 4 : 0 |
| ≥20 | 6 | 0 | 6 : 0 | 4 | 0 | 4 : 0 |

Note –The ratio of $A_i / S_i$ is provided as a measure of selection strength.

# Results

In **PART I**, we search for the general mutation characteristics at and near high recurrence sites by both machine learning and extensive sequence comparisons. In **PART II**, we develop the mathematical theory for the maximal level of recurrences of neutral mutations (designated $i^*$). CDNs are thus defined as mutations with $\geq i^*$ recurrences in **n** cancer samples. We then expand in **PART III** the theory to very large sample sizes ($n \geq 10^5$), thus making it possible to identify all CDNs.

To carry out the analyses, we first compile from the TCGA database the statistics of multiple hit sites ($i$ hits in **n** samples) in 12 cancer types. This study focuses on the mutational characteristics pertaining to recurrence sites. We often present the three cancer types with the largest sample sizes (lung, breast, and CNS cancers), while many analyses are based on pan-cancer data. Analyses of every of the 12 cancer types individually will be done in the companion paper. The TCGA database is used as it is well established and covers the entire coding region (*Cancer Genome Atlas Research Network et al., 2013*). Other larger databases (*Cerami et al., 2012*; *Tate et al., 2019*; *de Bruijn et al., 2023*) are employed when the whole exon analyses are not crucial.

## The compilation of multi-hit sites across all genes in the genome

Throughout the study, $S_i$ denotes the number of synonymous sites where the same nucleotide mutation occurs in $i$ samples among **n** patients. $A_i$ is the equivalent of $S_i$ for non-synonymous (amino acid altering) sites. *Table 1* presents the numbers from lung cancer for demonstration. It also shows the $A_i$ and $S_i$ numbers with CpG sites filtered out. There are 22.5 million nonsynonymous sites, among which ~0.2 million sites have one hit ($A_1$=195,958) in 1035 patients. The number then decreases sharply as $i$ increases. Thus, $A_2$=2946 (number of 2-hit sites), $A_3$=99, and $A_4 + A_5 + \ldots = 79$. We also note that the $A_i/S_i$ ratio increases from 2.89, 2.82, 3.04–4.71 and so on.

*Figure 2* shows the average of $A_i$ and $S_i$ among the 12 cancer types (see Methods). The salient features are shown by differences between the solid and dotted lines. As will be detailed in **PART II**, the dotted lines, extending linearly from $i$=0 to $i$=1 in logarithmic scale, should be the expected values of $A_i$ and $S_i$, if mutation rate is the sole driving force. In the actual data, ($A_1$) and $S_1$ decrease to ~0.002 of $A_0$ and $S_0$, the step being least affected by selection (see **PART II** later). For $A_2$ and $S_2$, the decrease is only ~0.01 of $A_1$ and $S_1$. The decrease from $i$ to $i$+1 becomes smaller and smaller as $i$ increases, suggesting that the process in not entirely neutral. Furthermore, the lower panel of *Figure 2* shows that $A_i/S_i$ continues to rise as $i$ increases. These patterns again suggest a stronger positive selection at higher $i$ values. The extrapolation lines shown in *Figure 2* roughly define $i$=3 as a cutoff where the expected ($A_3$) falls below 1 (see **PART II** for details). The precise model of **PART II** will define high recurrence sites ($i \geq 3$) as CDNs.

## PART I - The mutational characteristic of high recurrence sites

In this part, the analyses are done in two different ways. The sequence-feature approach is to examine the mutation characteristics of sequence features (say, 3 mers, 5 mers, etc.) across patients. The patient-feature approach is to examine patients for their mutation signatures and mutation loads.

### The sequence-feature approach

The simplest and best-known sequence feature associated with high mutation rate is CpG sites. In mammals, methylation and de-amination would enhance the mutation rate from C̲pG to T̲pG or Cp̲A by five-~tenfold (*Hodgkinson and Eyre-Walker, 2011*; *Ségurel et al., 2014*). As the CpG site mutagenesis has been extensively reported, we only present the confirmation in the Supplement (*Figure 2—figure supplement 1*). Indeed, CpG sites account for ~6.5% of the coding sequences but contributing ~22% among the mutated sites in *Figure 2*. Hence, the sevenfold increase in the CpG mutation rate should contribute more to $A_i$ and $S_i$ as $i$ increases. *Table 1* has shown the effects of filtering out CpG sites in the counts of recurrences. Clearly, CpG sites do contribute disproportionately to the recurrences but, even when they are separately analyzed, the conclusion is unchanged. As shown later in **PART II**, every increment of $i$ should decrease the site number by ~0.002 in the TCGA database. Thus, even with a 10-fold increase in mutation rate, the decrease rate would still be 0.02. In the theory sections, CpG site mutations are incorporated into the model.

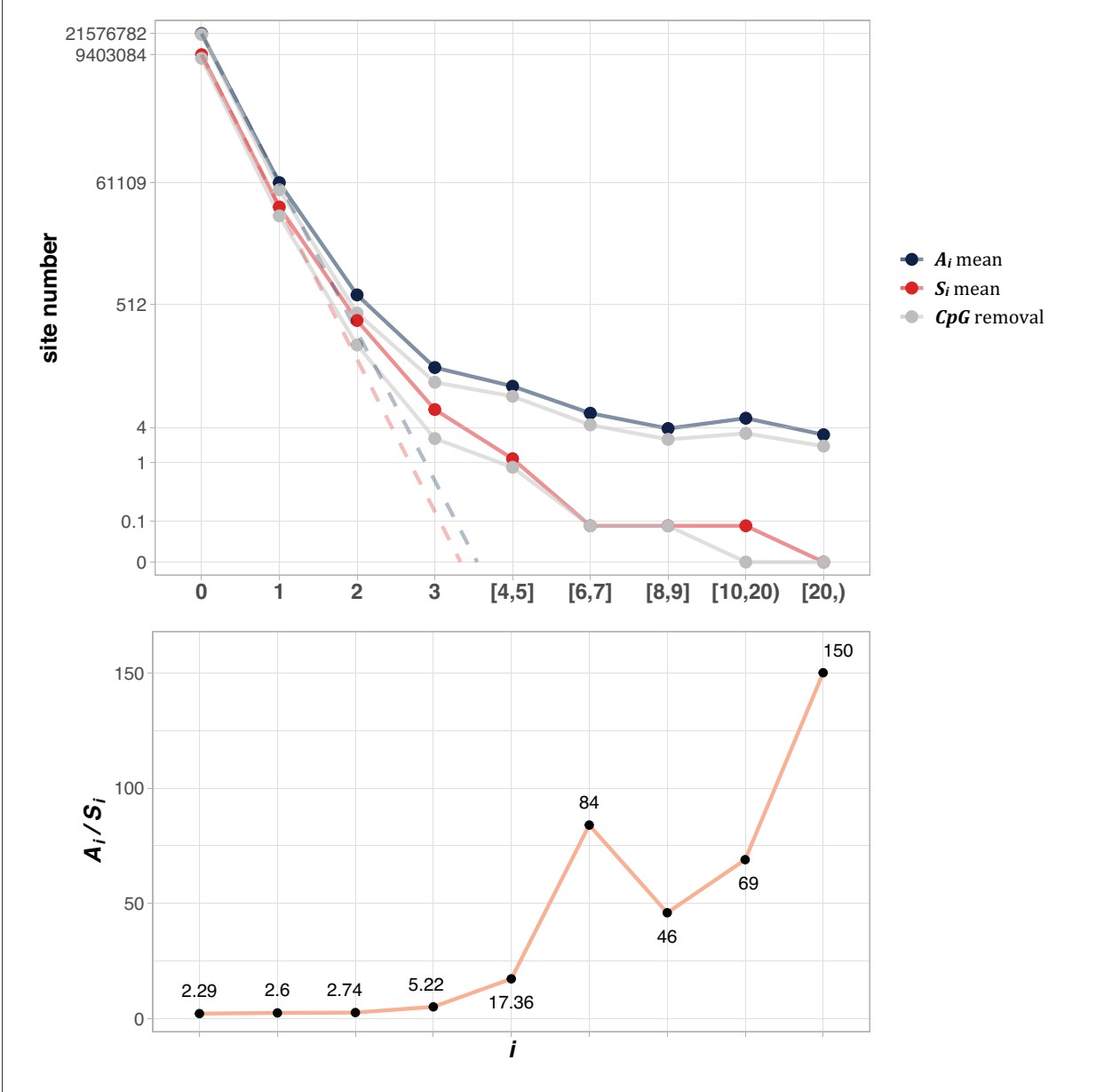

**Figure 2.** The average $Ai$ and $Si$ values across different $i$ ranges (X-axis). (Top): The average of $Ai$ and $Si$ in the log scale. Color lines - full data; gray lines - CpG sites removed. The dash lines are linear extrapolations. **Bottom:** The $A_i/S_i$ ratio as a function of $i$. The drop of $A_i/S_i$ ratio at $i$ [8, 9] is due to the potential synonymous CDNs, see ***Supplementary file 1***.

The online version of this article includes the following figure supplement(s) for figure 2:

**Figure supplement 1.** Mutation and context landscape across 12 cancer types.

In this section, we aim to find out how extreme the mutation mechanisms must be to yield the observed recurrences. If these mechanisms seem implausible, we may reject the mutational-hotspot hypothesis and proceed to test the functional hotspot hypothesis.

## The analyses of mutability variation by Artificial Intelligence (AI)

The variation of mutation rate at site level could be shaped by multiple mutational characteristics. Epigenomic features, such as chromatin structure and accessibility, could affect regional mutation rate at kilobase or even megabase scale (***Stamatoyannopoulos et al., 2009***; ***Makova and Hardison, 2015***), while nucleotide biases by mutational processes typically span only a few base pairs around

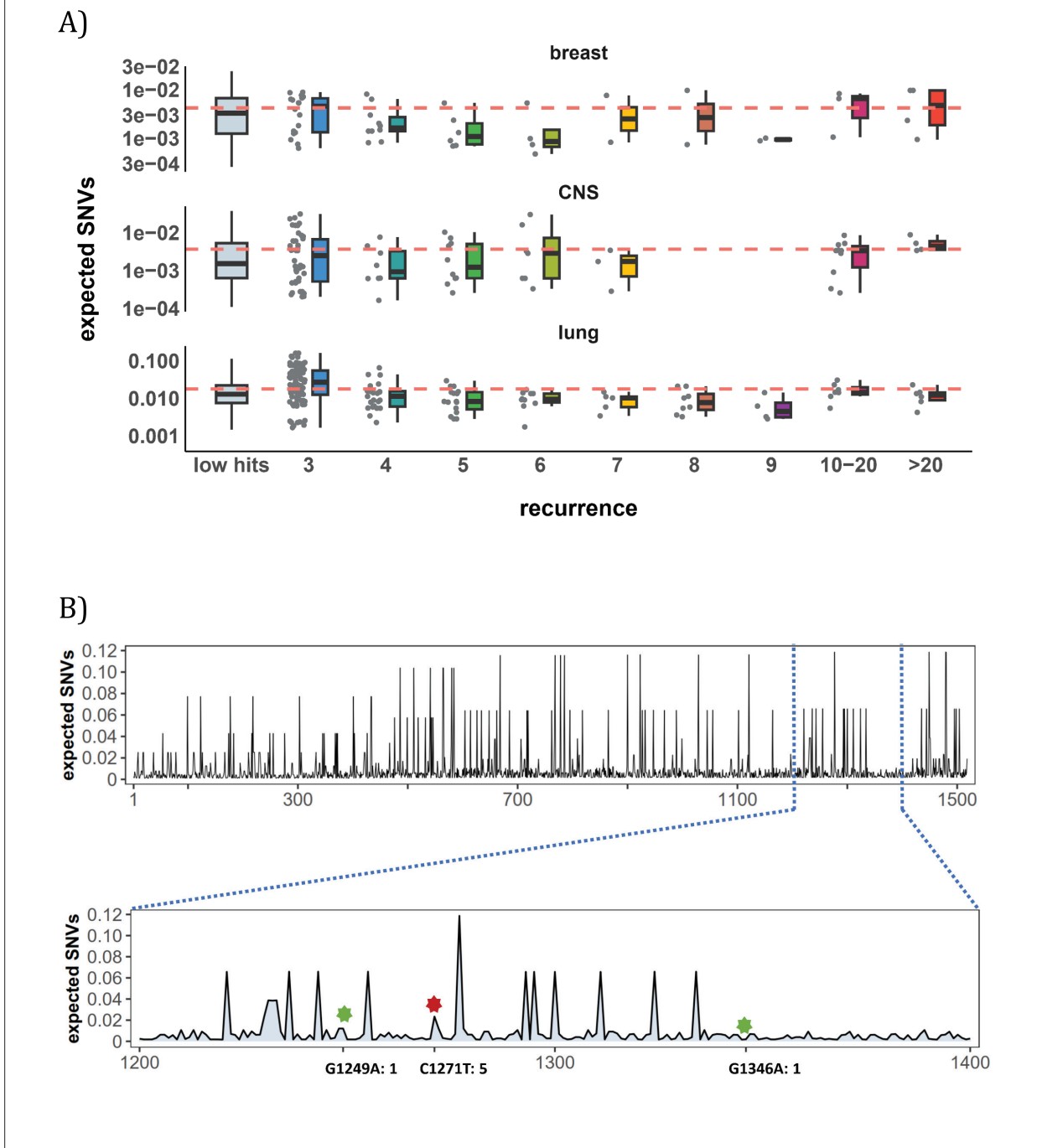

**Figure 3.** Site-level mutation rate variation obtained from *Dig Sherman et al., 2022*, a published AI tool. (**A**) Each dot represents the expected SNVs (Y-axis) at a site where missense mutations occurred *i* times in the corresponding cancer population. The boxplot shows the overall distribution of mutability at *i*, with the red dashed line denoting the average. There is no observable trend that sites of higher *i* are more mutable (The blank areas are due to the absence of CDNs with mutation recurrence counts of 8 or 9 in CNS cancer mutation data, see *Supplementary file 1*). (**B**) A detailed look at the coding region of *PAX3* gene in colon cancer. The expected mutability of sites in the 200 bp window is plotted. The three mutated sites in this window, marked by green and red (a CDN site) stars, are not particularly mutable. Overall, the mutation rate varies by about tenfold as is generally known for CpG sites.

the mutated site (*Roberts et al., 2013*; *Haradhvala et al., 2018*; *Herzog et al., 2021*). AI-powered multi-modal integration offers a new tool to quantify the joint effect of various factors on mutation rate variability (*Luo et al., 2019*; *Sherman et al., 2022*; *Song et al., 2023*). Here we explore the association between the mutation recurrence (*i*) and site-level mutation rate predicted by AI.

*Figure 3A* shows the mutation rate landscape across all recurrence sites in breast, CNS and lung cancer using the deep learning framework *Dig* (*Sherman et al., 2022*). In this approach, the mutability of a focus site is calculated based on both local stretch of DNA and broader scale of epigenetic features. The X-axis shows all mutated sites with $i>0$ scanned by *Dig*. While the mutation rate fluctuates around the average level, we detect no significant difference in mutation rate as a function of $i$ (Methods). In CNS, two sites exhibited exceptional mutability at $i=6$, surpassing the average by tenfold. Unsurprisingly, these two are CpG sites and correspond to amino acid change of V774M and R222C in *EGFR* (*Supplementary file 1*), which are canonical actionable driver mutations in glioma target therapy. In other words, the two sites called by AI for possible high mutability appear to be selection driven.

In *Figure 3B*, we take a closer look at how CDN is situated against the background of mutation rate variation, using the example of *PAX3* (Paired Box Homeotic Gene 3; *Wang et al., 2008*; *Li et al., 2019*). In this typical example, *Dig* predicts site mutability to vary from site to site. In the lower panel is an expanded look at a stretch of 200 bps. In this stretch, about 8% of sites are five- to tenfold more mutable than the average. But none of them are mutated in the data ($i=0$). There are indeed three sites with $i>0$ in this DNA segment including a CDN site C1271T (marked by the red star). This CDN site is estimated to have a twofold elevation in mutability, which is less than 1/50 of the necessary mutability to reach $i\geq3$. The other two mutated sites, marked by the green star, are also indicated.

Other AI methods have also been used in the mutability analysis (*Fang et al., 2022*), reaching nearly the same conclusion. Overall, while AI often suggests sequence context to influence the local variation in mutation rate, the reported variation does not correspond to the distribution of CDNs. In the next subsection, we further explore the local contexts for potential biases in mutability.

## The conventional analyses of local contexts - from 3-mers to 101-mers

Since the AI analyses suggest the dominant role of local sequence context in mutability, we carry out such conventional analyses in depth. Other than the CpG sites, local features such as the TCW (W=A or T) motif recognized by APOBEC family of cytidine deaminases (*Burns et al., 2013*; *Roberts et al., 2013*), would have impacts as well. We first calculate the mutation rate for motifs of 3-mer, 5-mer and 7-mer, respectively, with 64, 1024, 16384 in number (see Methods). The pan-cancer analyses across

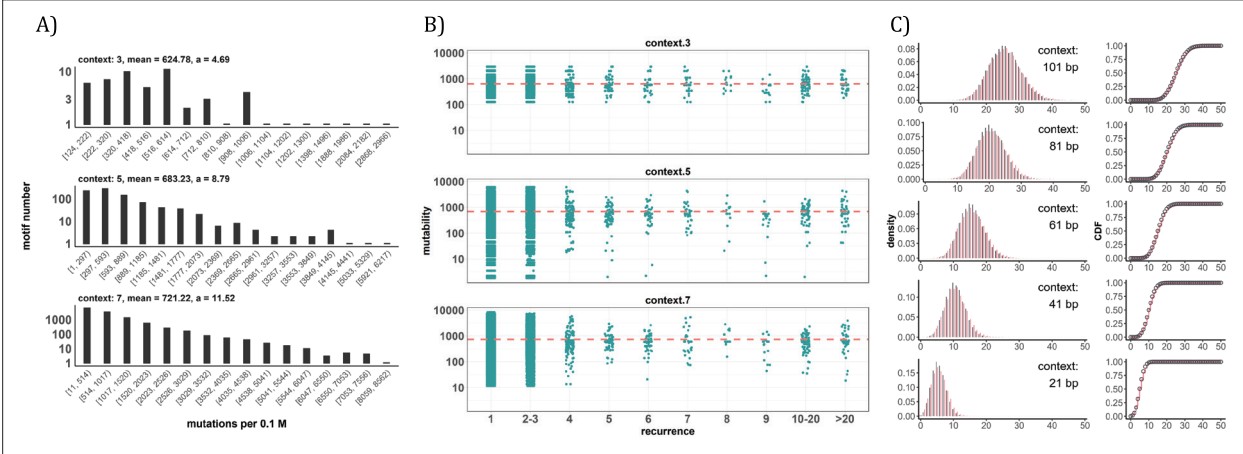

**Figure 4.** Conventional analyses of local contexts at recurrence sites. (**A**) From top panel down - For the 64 ($4^3$) 3-mer motifs, their mutational rates are shown on the X-axis. The most mutable motif over the average mutability ($\alpha$) is 4.69. For the 1024 (=$4^5$) 5-mer and 16,384 ($4^7$) 7-mer motifs, the $\alpha$ values are, respectively, 8.79 and 11.52. The most mutable motifs, as expected, are dominated by CpG's. (**B**) Each dot represents the motif surrounding a high-recurrence site. The recurrence number is shown on the X-axis and the mutability of the associated motif's mutability (mutations per 0.1 M) is shown on the Y-axis. The average mutation rate across all motifs of given length category is indicated by a red horizontal dashed line. The absence of a trend indicates that the high recurrence sites are not associated with the mutability of the motif. (**C**) The analysis is extended to longer motifs surrounding each CDN (21, 41, 61, 81, and 101 bp). For each length group, all pairwise comparisons are enumerated. The observed distributions (black bars and points) are compared to the expected Poisson distributions (red bars and curves) and no difference is observed. Thus, local sequences of CDNs do not show higher-than-expected similarity.

The online version of this article includes the following figure supplement(s) for figure 4:

**Figure supplement 1.** Sliding window to explore the consensus sequences between recurrence sites.

the 12 cancer types are shown in *Figure 4A*. We use α to designate the fold change between the most mutable motif and the average. Since the number of motifs increases 16-fold between each length class, the α value increases from 4.7 to 8.8 and then to 11.5. Nevertheless, even the most mutable 7-mer, TAA<u>C</u>GCG, which has a CpG site at the center, is only 11.52-fold higher than the average. This spread is insufficient to account for the high recurrences, which decrease to ~0.002 for each increment of $i$ (see **PART II** below).

A more direct approach is given in *Figure 4B* explained in the legends. The absence of a trend shows that the high recurrence sites are not associated with the mutability of the motif. In *Figure 4C*, the analysis extends to longer motifs of 21 bp, 41 bp, 61 bp, 81 bp, and 101 bp surrounding the high-recurrence sites. For example, the motif of 101 bp may be (10, 90), (20, 80) and so on either side of a recurrence site (*Figure 4—figure supplement 1*). We then compute the pairwise differences in sequences of the motifs among recurrence sites. The logic is that, if certain motifs dictate high mutation rates, we may observe unusually high sequence similarity in the pairwise comparisons. As can be seen in all 5 panels of *Figure 4C*, there are no outliers in the tail of the distribution. In other words, the sequences surrounding the high-recurrence sites appear rather random. Detailed motif analysis of CDNs within individual cancer types using deep learning models (ResNet, LSTM and GRU) further supports this conclusion.

In conclusion, the analyses by the sequence approach do not find any association between high recurrence sites ($i \geq 3$) and the mutability of the local sequences.

## The patient-feature approach

In this second approach, we examine the mutation characteristics among patients across sequence features. The first question is whether high recurrence sites tend to happen in patients with higher mutation loads. *Figure 5A* depicts the distribution of mutation loads among patients harboring a CDN of recurrence $i$. Hence, a patient's load may appear several times in the plot, each appearance corresponding to one CDN in the patient's data. For the comparison across $i$ values, the mutation load is normalized by a z-score within each cancer population to equalize the three cancer types. The overall trend shows consistently that patients with recurrence sites do not bias toward high mutation loads. The presence of recurrence sites in patients with low mutation loads suggests that overall mutation burden is not a determining factor of recurrence.

With the results of *Figure 5A*, we then ask a related question: whether these high recurrence mutations are driven by factors that affect mutation characteristics. Such influences have been captured by the analyses of 'mutational signatures' (*Alexandrov et al., 2013*; *Alexandrov et al., 2020*). Each signature represents a distinct mutation pattern (e.g. high rate of T<u>C</u>T ->T<u>A</u>T and other tri-nucleotide changes) associated with a known factor, such as smoking or an aberrant mutator (aristolochic acid, for example). Each patient's mutation profile can then be summarized by the composite of multiple mutational signatures.

The issue is thus whether a patient's CDNs can be explained by the patient's composite mutational signatures. *Figure 5B* reveals that in lung cancer, the signature compositions among patients with different recurrence cutoffs are statistically indistinguishable (Methods). Smoking (signature SBS4) consistently emerges as the predominant mutational process across all levels of recurrences. In breast cancer, while SBS2 and SBS13 exhibit some differences in the bins of $i^*=2$ and $i^*=3$, the profiles remain rather constant for all bins of $i^* \geq 3$. The two lowest bins, not unexpectedly, are also different from the rest in the total mutation load (see *Figure 5A*). In *Appendix 1—table 1*, we provide a comprehensive review of supporting literature on genes with recurrence sites of $i \geq 3$ for breast cancer. In CNS cancer, SBS11 appears significantly different across bins, in particular, $i \geq 20$. This is a signature associated with Temozolomide treatment and should be considered a secondary effect. In short, while there are occasional differences in mutational signatures across $i^*$ bins, none of such differences can account for the recurrences (see **PART II**).

To conclude **PART I**, the high-recurrence sites do not stand out for their mutation characteristics. Therefore, the variation in mutation rate across the whole genome can reasonably be approximated mathematically by a continuous distribution, as will be done below.

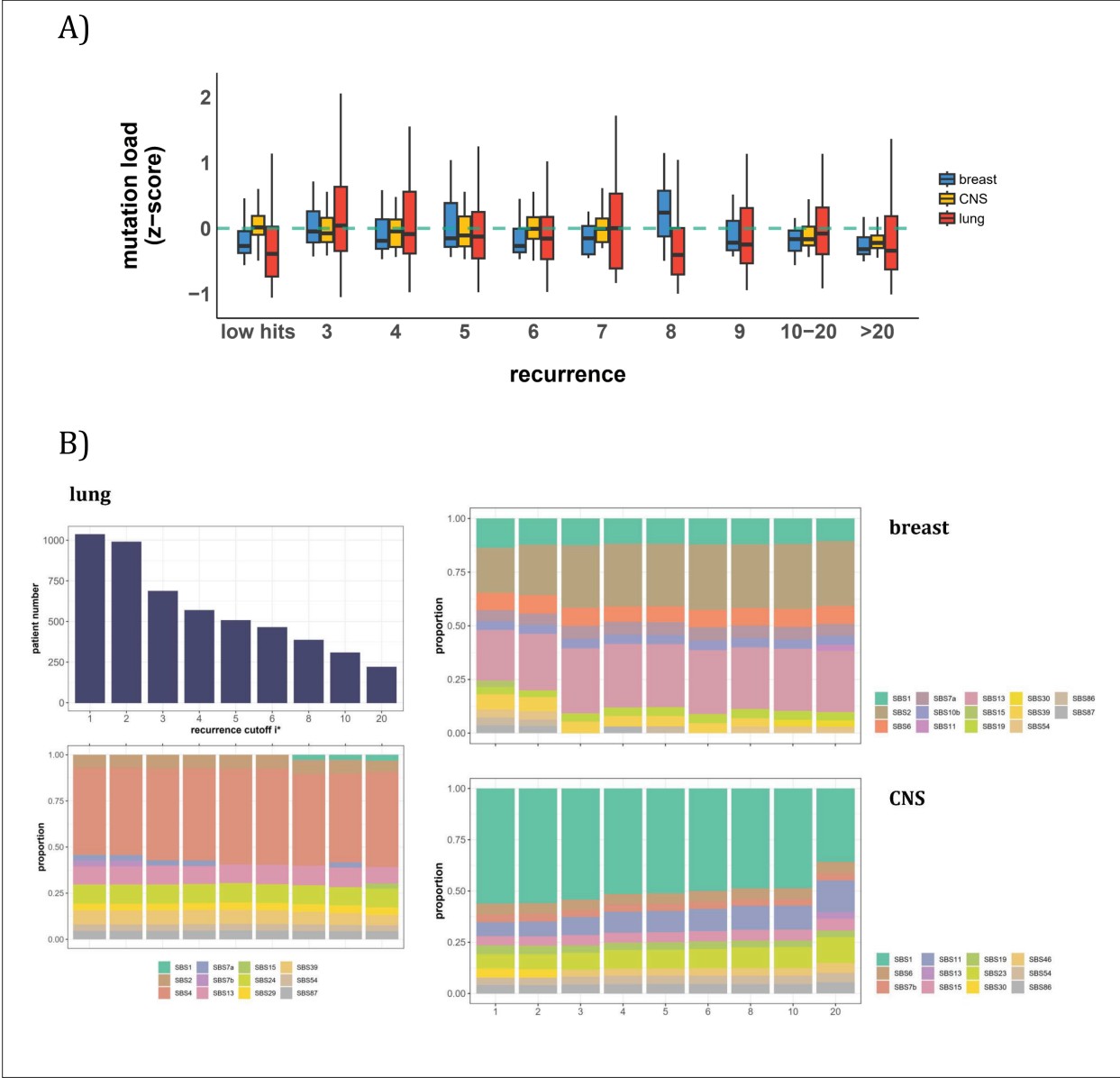

**Figure 5.** Patient level analysis for mutation load and mutational signatures. (**A**) Boxplot depicting the distribution of mutation load among patients with recurrent mutations. The X-axis denotes the count of recurrent mutations, while the Y-axis depicts the normalized z-score of mutation load (see Methods). The green dashed line indicates the mean mutation load. In short, the mutation load does not influence the mutation recurrence among patients. (**B**) Signature analysis in patients with mutations of recurrences $\geq i^*$ (X-axis). For lung cancer (left), the upper panel presents the number of patients for each group, while the lower panel depicts the relative contribution of mutational signatures. For breast cancer, APOBEC-related signatures (SBS2 and SBS13) are notably elevated in all groups of patients with $i^* \geq 3$, while patients with mutations of recurrence $\geq 20$ in CNS cancer exhibit an increased exposure to SBS11 (*Blough et al., 2011*; *Lin et al., 2021*; *Noeuveglise et al., 2023*). Again, patients with higher mutation recurrences do not differ in their mutation signatures.

## PART II - The theory of CDNs

We now develop the theory for $S_i$ and $A_i$ under neutrality where $i$ is the recurrence of the mutation at each site. We investigate the maximal level of neutral mutation recurrences ($i^*$), above which the expected values of $S_i$ and $A_i$ are both <0. Since no neutral mutations are expected to reach $i^*$, every mutation with the recurrence of $i^*$ or larger should be non-neutral. Importantly, given that the expected $S_i$ and $A_i$ is a function of $U^i$ where $U=nE(u)$ is in the order of $10^{-2}$ and $10^{-3}$, $i^*$ is insensitive to a wide range of mutation scenarios. For that reason, the conclusion is robust.

The mutation rate of each nucleotide ($u$) follows a Gamma distribution with a scale parameter $\theta$ and a shape parameter $k$. Gamma distribution is often used for its flexibility and, in this context, models the waiting time required to accumulate $k$ mutations. Its mean ($=k\theta$) and variance ($=k\theta2$) are determined by both parameters but the shape (skewness and kurtosis) is determined only by $k$. In particular, the Gamma distribution has a long tail suited to modeling a small number of sites with very high mutation rate.

We now use synonymous ($S_i$) mutations as the proxy for neutrality. Hence, in **n** samples,

$$S_i = \sum_{l=1}^{L_S} C_n^i u\,(l)^i \left[1 - u\,(l)\right]^{n-i} \sim C_n^i L_S E\left(u^i\right) \tag{1}$$

where $L_S$ is the total number of synonymous sites and $u(l)$ is the mutation rate of site $l$. In the equation above, the term $\left[1 - u\,(l)\right]^{n-i}$ is dropped. We note that $\left[1 - u\,(l)\right]^{n-i} e^{\left[-u(l)\,(n-i)\right]} \sim 1$ as $u$ is in the order of $10^{-6}$ and $n$ is in the order of $10^2$ from the TCGA data. With the gamma distribution of $u$ whereby the $i^{\text{th}}$ moment is given by

$$E\left(u^i\right) = \frac{\Gamma\,(k+i)}{\Gamma\,(k)} \theta^i = \frac{\Gamma\,(k+i)}{\Gamma\,(k) \cdot k^i} E\,(u)^i$$

we obtain:

$$S_i = L_S \cdot g\,(i, k) \left[nE\,(u)\right]^i$$

here:

$$g\,(i, k) = C_{k+i-1}^i \cdot \frac{1}{k^i}$$

In a condensed form,

$$S_i = G \cdot \left[nE\,(u)\right]^i \tag{2}$$

where $G = L_S \cdot g\,(i, k)$. Similarly, if nonsynonymous mutations are assumed neutral, then,

$$A_i = L_A \cdot g\,(i, k) \left[nE\,(u)\right]^i \sim 2.3 \cdot S_i \tag{3}$$

The number of 2.3 is roughly the ratio of the total number of nonsynonymous over that of synonymous sites (*Hartl and Clark, 1989*; *Li, 1997*; *Chen et al., 2019*). This number would vary moderately among cancers depending on their nucleotide substitution patterns.

**E(u)** of *Equations 2 and 3* is generally $(1\sim5)\times10^{-6}$ per site in cancer genomic data and **n** is generally between 300 and 1000. Hence, **nE(u)** is the total mutation rate summed over all **n** patients and is generally between 0.001 and 0.005 in the TCGA data. Given **nE(u)** is in the order of $10^{-3}$, $S_i$ and $A_i$ would both decrease by 2~3 orders of magnitude with each increment of $i$ by 1. We note that the total number of synonymous sites, $L_S$, is $\sim0.9 \times 10^7$ and $L_A$ is ~2.3 times larger. Therefore, $S_3 <1$ and $A_3 <1$. When $i$ reaches 4, $S_{i\,\geq4}$ and $A_{i\,\geq4}$ would both be $\ll 1$ when averaged over cancer types.

For each cancer type, the conclusion of $S_3 <1$ and $A_3 <1$ is valid with the actual value of $S_3$ and $A_3$ ranging between 0.01 and 1. In other words, with **n**<1000,, neutral mutations are unlikely to recur 3 times or more in the TCGA data ($i^*$=3). While $S_{i\,\geq3}$ and $A_{i\,\geq3}$ sites are high-confidence CDNs, the value of $i^*$ is a function of **n**. At **n**≤1,000 for the TCGA data, $i^*$ should be 3 but, when **n** reaches 10,000, $i^*$ will be 6. The benefits of large **n**'s will be explored in **PART III**.

## Possible outliers to the distribution of mutation rate

Although we have explored extensively the sequence contexts, other features beyond DNA sequences could still lead to outliers to mathematically distributions. These features may include DNA stem loops (*Buisson et al., 2019*) or unusual epigenetic features (*Zheng et al., 2014*; *Makova and Hardison, 2015*; *Supek and Lehner, 2015*). We therefore expand the model by assuming a small fraction of sites ($p$) to be hyper-mutable that is $\alpha$ fold more mutable than the genomic mean. Most likely, $\alpha$ and $p$ are the inverse of each other. For the bulk of sites (1 $p$) of the genome, we assume that their mutations

**Table 2.** Summary for modeling outlier sites in six cancer types.

| Cancer Type | $S_3$ | p | α | $S_4$ | $S_5$ |
|---|---|---|---|---|---|
| Lung* | -- | 0.0 | -- | -- | -- |
| Breast | 0.12 | 8.75E-04 (8.21E-04) | 88.6 (32.0) | 0.102 (0.068) | 0.004 (0.004) |
| CNS | 0.02 | 2.73E-04 (1.09E-04) | 295.1 (57.0) | 0.448 (0.173) | 0.026 (0.015) |
| Kidney | 0.03 | 3.03E-05 (2.98E-05) | 304.1 (108.0) | 0.067 (0.056) | 0.005 (0.006) |
| Upper-AD tract | 0.47 | 0.002 (0.001) | 48.9 (10.7) | 0.174 (0.078) | 0.005 (0.003) |
| Large intestine | 1.03 | 0.009 (0.001) | 51.6 (1.4) | 0.998 (0.087) | 0.026 (0.003) |

Note – For each cancer type, p stands for the proportion of highly mutable sites, with mutation rate being α-fold of the average. $S_3$ gives the expected number without mutable outliers (P=0). $S_4$ and $S_5$ denote the expected number with the best (p, α) pairs with the standard deviation in parentheses. For lung cancer, $S_2$ and $S_3$ do not fit the outlier model (**Table 2—source data 1**); therefore, we set P=0.

The online version of this article includes the following source data for table 2:

**Source data 1.** The outlier model parameters and expected Si values for 6 cancer types analyzed.

follow the Gamma distribution. (Nevertheless, the bulk can be assumed to have a fixed mutation rate of $E(u)$ without affecting the conclusion qualitatively.)

We let p range from $10^{-2}$ to $10^{-5}$ and α up to 1000. As no stretches of DNA show such unusually high mutation rate (1000-fold higher than the average), such sites are assumed to be scattered across the genome and are rare. With the parameter space of defined above, we choose the (p, α) pairs that agree with the observed values of $S_1$ to $S_3$ which are sufficiently large for estimations. **Table 2** presents the value range and standard deviation for p and α across the six cancer types that have >500 patient samples. Among the six cancer types, the lung cancer data do not conform to the constraints and we set p=0. With observed values for $S_{1-3}$ as constraints, $S_4$ rarely exceeds 1. Hence, even with the purely conjectured existence of outliers in mutation rate, $i^*$=4 is already too high a cutoff.

**Table 2** suggests that p has to be smaller than $10^{-5}$ and α>1,000 to yield $S_4$ >1. Since the coding region has $3×10^7$ sites, p<$10^{-5}$ would mean that the outliers are at most in the low hundreds. In other words, the number of high recurrence sites projected by the theory is close to the observed numbers. Therefore, there are really no unknown outlier sites of high mutation rate. Positive selection would be a more straightforward explanation, explained below.

## The influence of selection on mutation occurrences

We now show that, although the mutational bias alone cannot account for the high occurrences, selection can easily do so. We assume a fraction, f, of $A_i$'s to be under positive selection. The fraction should be small, probably ≪ 0.01, and will be labeled $A_i^*$. The rest, labeled $A_i$ is considered neutral. Hence, $A_i$ is proportional to $[nE(u)]^i$ and $A_i/S_i = L_A/L_S$ ~2.3. Like $S_{i \geq 4}$, $A_{i \geq 4}$ ≪ 1. In contrast,

$$A_i^* = G^*[w \cdot nE(u)]^i \tag{4}$$

$$\frac{A_i^*}{S_i} = \frac{G^*}{G}w^i \tag{5}$$

where $G^*$ is also a constant but its value depends on f. The crucial parameter, $w$ (=$2Ns$), is the selective advantage (s) scaled by the population size of progenitor cancer cell (N). Since $w$ can easily be >10, even at i=3, $A_1^*/S_1$ would be >100 as large as $(1-u)^{n-i}e^{-u(n-i)}$. In other words, observed mutation recurrences at i≥3 for advantageous mutation should not be uncommon. **Equation 4** also shows that $w$ and $E(u)$ jointly affect the recurrence; therefore, CpG sites (many of which fall in functional sites) are expected to be strongly represented among high recurrence sites.

## PART III. The theory of large samples (n > $10^5$) and identification of all CDNs

Using the theory of CDN developed above, the companion paper shows that each sequenced cancer genome in the current databases, on average, harbors only 1~2 CDNs. The number varies in this range depending on the cancer type (**Zhang et al., 2024**). For comparison, tumorigenesis may require at

least 5~10 driver mutations as estimated by various criteria (*Armitage and Doll, 1954*; *Hanahan and Weinberg, 2011*; *Belikov, 2017*; *Martincorena et al., 2017*; *Anandakrishnan et al., 2019*; *ICGC/ TCGA Pan-Cancer Analysis of Whole Genomes Consortium, 2020*). The results show that there are many more CDNs that have not been discovered. This is not unexpected since most CDNs are found in <1% of patients. If each CDN is observed in 1% of patients and each patient has 5 CDNs, then there should be at least 500 CDNs for each cancer type.

In the companion study that uses the *A/S* ratios of *Figure 1*, the estimated number of CDNs ranges from 500 to 2000, whereas the current estimates based on $A_{i \geq 3}$ sites is only 50~100 (*Zhang et al., 2024*). Where, then, are the undiscovered CDNs and how to find them? Since all $A_{i \geq 3}$ sites are concluded to be CDNs, the bulk of CDNs must be among $A_1$ and $A_2$ sites. The best way to identify the CDNs hidden in $A_1$ and $A_2$ is to increase the sample size, **n**, dramatically.

We hence extend *Equation 1* for $S_i$ and $A_i$ to large **n**'s. Note that *Equation 1* drops the term of $(1 - u)^{n-i} \sim e^{-u(n-i)}$ as it is ~1, when **nE(u)** ≪ 1. With a large **n** when $e^{-u(n-i)}$ is not near 1, the recurrence of mutations would follow a Poisson distribution with the expected value of **nE(u)**. Assuming that *u* follows a gamma distribution with a shape parameter of *k*, the probability of observing *i* mutations would follow the negative binomial distribution as shown below:

$$f\left(i|k, n, E\left(u\right)\right) = \frac{\Gamma\left(i+k\right)}{\Gamma\left(i+1\right)\Gamma\left(k\right)} k^k \left[nE\left(u\right)\right]^i \left[k + nE\left(u\right)\right]^{-k-i} \tag{6}$$

The cumulation density function for *Equation 6* is then:

$$F\left(i \leq i^*\right) = \sum_{i=1}^{i^*} f\left(i|k, n, E\left(u\right)\right) \tag{7}$$

Then, by definition, $A_{i \geq i^*}$ should be ≪ 1 so that mutations with recurrences $i>i^*$ could be defined as CDN. Thus:

$$F\left(i \leq i^*\right) = 1 - \frac{\varepsilon}{L_A} \tag{8}$$

where $\varepsilon = A_{i \geq i^*}$ denotes the number of sites with mutation recurrence $\geq i^*$ under the sole influence of mutational force. $\frac{\varepsilon}{L_A}$ could then be regarded as significance of $i^*$ since it controls the overall false positive rate of CDNs.

Specifically, with *k*=1, the probability function of mutation recurrence of a given site would transform to a geometric distribution with $P$=1 / (1+**nE(u)**), the cumulative density function (CDF) is then:

$$F\left(i \leq i^*\right) = 1 - \left(1 - \frac{1}{1 + nE\left(u\right)}\right)^{i^*} \tag{9}$$

Combined with *Equation 5*, the mutation recurrence cutoff $i^*$ of being a CDN could be expressed as:

$$i^* = \frac{\log\left(\frac{\varepsilon}{L_A}\right)}{\log\left(\frac{nE\left(u\right)}{1 + nE\left(u\right)}\right)} \tag{10}$$

For very large **n**, 1/**nE(u)** is small and $i^*$/**n** can be approximated as

$$i^*/n = \log\left(L_A\right) \cdot E\left(u\right) \sim 5 \times 10^{-5} \tag{11}$$

*Equation 11* shows that $i^*$/**n** would approach asymptotically as **n** increases. This asymptotic value is attained when n reaches ~$10^6$.

*Figure 6* shows the range of $i^*$ for **n** up to $10^6$. As expected, $i^*$ increases by small increments while **n** increases in 10-fold jumps. For example, when **n** increases by 3 orders of magnitude, from 100 to 100,000,, $i^*$ only doubles from 3 to 12. The disproportional increment between $i^*$ and **n** explains why

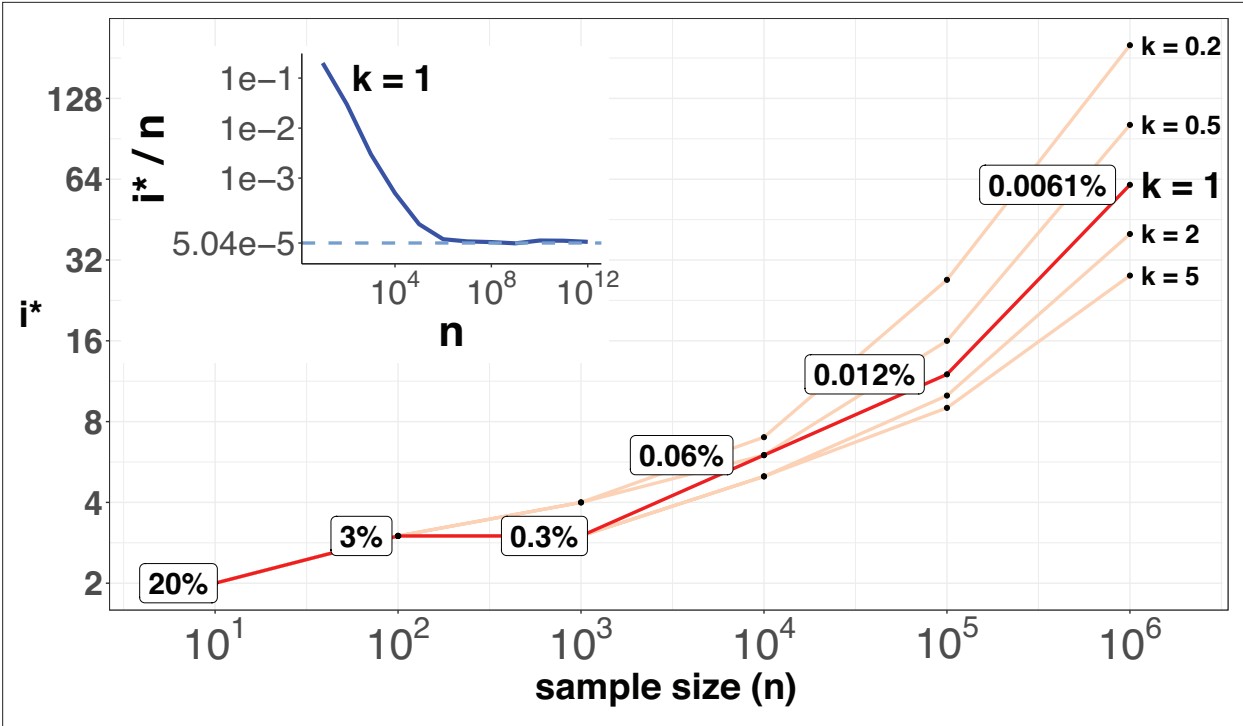

**Figure 6.** $i^*$ values (Y-axis, log scale) against sample sizes (n), X-axis across different shape parameter $k$'s. The Y axis presents the $i^*$ values under different sample sizes (n) of the X-axis in log scale. Five shape parameters (k) of the gamma-Poisson model are used. In the literatures on the evolution of mutation rate, k is usually greater than 1. The inset figure illustrates how $i^*/n$ (prevalence) would decrease with increasing sample sizes. The prevalence would approach the asymptotic line of $[g(i,k)]$ when n reaches $10^6$. In short, more CDNs (those with lower prevalence) will be discovered as n increases. Beyond n=$10^6$, there will be no gain.

we use the actual number of $i^*$ for the cutoff, instead of the ratio $i^*/$**n**. As shown in the inset of **Figure 2**, the ratio of $i^*/$**n** would approach the asymptote at **n**~$10^6$, where an advantageous mutation only needs to rise to 0.00006 to be detected. With **n** reaching this level, we shall be able to separate most CDNs apart from the mutation background.

When **n** approaches $10^5$, the number of CDNs will likely increase more than 10-fold as conjectured in **Figure 7A**. In that case, every patient would have, say, 5 CDNs that can be subjected to gene targeting. (The companion study shows that, at present, an average cancer patient would have fewer than one targetable CDN.) Before the project is realized, it is nevertheless possible to test some aspects of it using the GENIE data of targeted sequencing. Such screening for mutations in the roughly 700 canonical genes serves the purpose of diagnosis with **n** ranging between 10~17 thousands for the breast, lung and CNS cancers. Clearly, GENIE efforts did not engage in discovering new mutations although they would discover additional CDNs in the canonical genes. In the companion study, we demonstrate that the analysis of CDNs identifies a potential set of 1.6 times more driver genes than those detected by whole gene selection signal calls (**Zhang et al., 2024**).

**Figure 7A** assumes that the prevalence, $i/$**n**, should not be much affected by **n**, but the cutoff for CDNs, $i^*/$**n**, would decrease rapidly as **n** increases. (**Figure 7B**) shows that the $i/$**n** ratios indeed correspond well between TCGA and GENIE, which differ by 10–20-fold in sample size. Importantly, as predicted in **Figure 6**, the number of CDNs increases by three to fivefold (**Figure 7C-E**). Many of these newly discovered CDNs from GENIE are found in the $A_1$ and $A_2$ classes of TCGA while many more are found in the $A_0$ class in TCGA. In conclusion, increasing **n** by one to two orders of magnitude would be the simplest means of finding all CDNs.

## Discussion

The nature of high-recurrence mutations has been controversial. Many authors have argued for mutational hotspots (**Hess et al., 2019**; **Stobbe et al., 2019**; **Nesta et al., 2021**; **Bergstrom et al., 2022**;

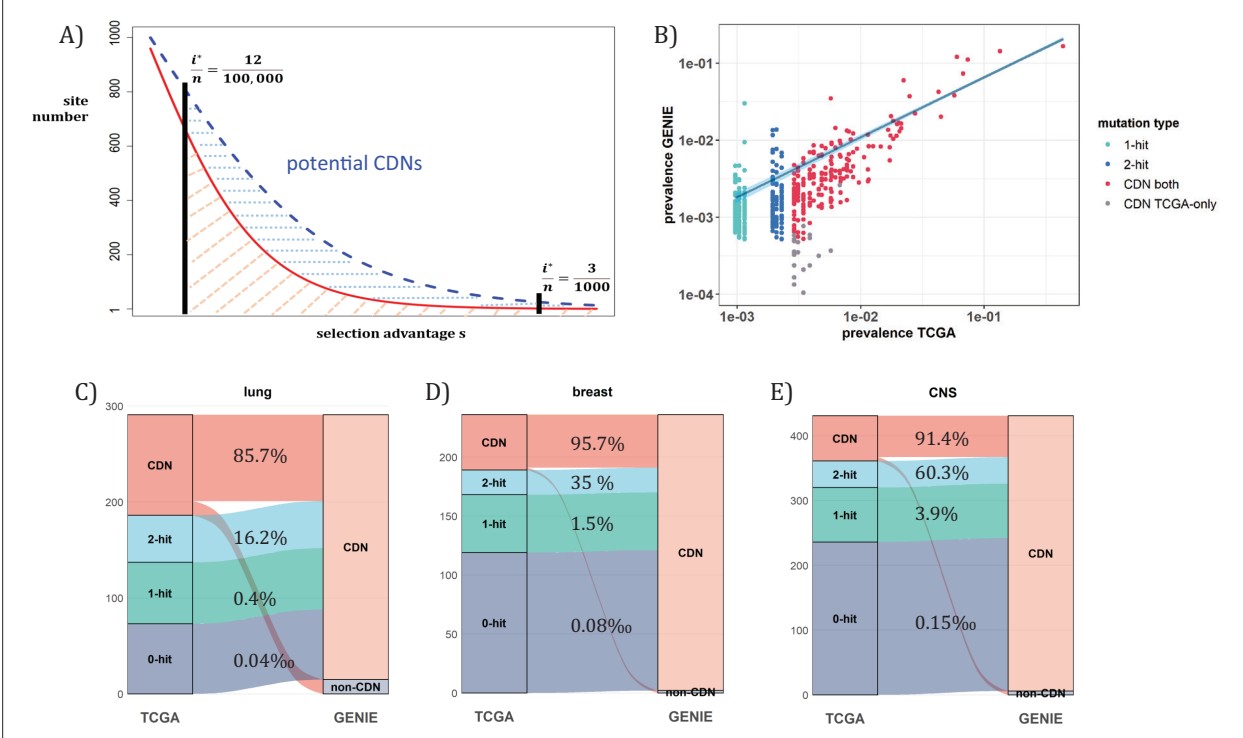

**Figure 7.** Analysis of CDNs with expanded sample set in GENIE. (**A**) Schematic illustrating the impact of sample size expansion on the number of discovered CDNs. The two vertical lines show the cutoffs of $i^*/n$ at (3/1000) vs. (12/100,000). The Y axis shows that the potential number of sites would decrease with $i^*/n$, which is a function of selective advantage. The area between the two cutoffs below the line represents the new CDNs to be discovered when n reaches 100,000. The power of n=100,000 is even larger if the distribution follows the blue dashed line. (**B**) The prevalence ($i/n$) of sites is well correlated between datasets of different **n** (TCGA with **n**<1000 and GENIE with generally tenfold higher), as it should be. Sites are displayed by color. '1-hit': CDNs identified in GENIE but remain in singleton in TCGA, '2-hit': CDNs identified in GENIE but present in doubleton in TCGA. 'CDN both': CDNs identified in both databases. (**C–E**) CDNs discovered in GENIE (n>9000) but absent in TCGA (n<1000). The newly discovered CDNs may fall in TCGA as 0–2 hit sites. The numbers in the middle column show the percentage of lower recurrence (non-CDN) sites in TCGA that are detected as CDNs in the GENIE database, which has much larger **n**'s.

Wong et al., 2022) but just as many have contended that they are CDNs driven by selection (*Gartner et al., 2013*; *Chang et al., 2016*; *Bailey et al., 2018*; *Cannataro et al., 2018*; *Juul et al., 2021*; *Zhao et al., 2021*; *Zeng and Bromberg, 2022*). While the two views co-exist, they are in fact incompatible. If the mutational hotspot hypothesis is correct, the selection hypothesis, and the determination of CDNs, would not be needed.

We believe that this study is the first to comprehensively test of the null hypothesis of mutational hotspots. In **PART I**, the mutational characteristics near all putative CDNs are examined and **PART II** presents the probability theory based on the analyses. The conclusion is that it is possible to reject the null hypothesis for recurrences as low as 3 in the TCGA data. The main reason for the high sensitivity is shown in *Equations 2 and 3* where $A_i$ and $S_i$ is proportional to $n_l$ or, roughly $0.002^i$. We recognized that the conclusion is based on what we currently know about mutation mechanisms. In a sense, the theory developed here can help the search for such unknown mutation mechanisms, if they do exist. Finally, the theory developed would permit the explorations in several new fronts when the sample size, **n**, expands to $10^5$.

The first front is to identify (nearly) all CDNs. When **n** reaches $10^5$, any point mutation with a prevalence higher than 12/100,000 would be a CDN, which is 25-fold more sensitive than in the TCGA data (3/1000). The companion analysis suggests that CDNs with lower prevalence, say 12/100,000 may still be highly tumorigenic in patients with the said mutation. If prevalence and potence are indeed poorly correlated, the search for lower prevalence CDNs by increasing **n** to $10^5$ is equivalent to searching for less common but still potent cancer driving mutations.

The second one is functional tests in patient-derived cell lines. When we have all CDNs identified, a patient can be expected to have multiple ($\geq$5; *Zhang et al., 2024*) CDNs. These mutations will be the

basis of in vitro test, as well as in animal model experiments, by gene editing, as shown recently (*Hodis et al., 2022*). Targeting multiple mutations simultaneously is necessary and may even be sufficient.

The third front, and arguably the most important one, is cancer treatment by targeted therapy (*Dang et al., 2017*; *Danesi et al., 2021*; *Waarts et al., 2022*; *Lin et al., 2023*; *Zhou et al., 2023*). When multiple CDN mutations in the same patient can be simultaneously targeted, the efficacy should be high. No less crucial, resistance to treatment should be diminished since it would be harder on cancer cells to evolve multiple escape routes to evade multiple drugs. Moreover, CDN analysis is crucial for stratifying patients for targeted therapy, as only targeting genes that are positively selected during cancer evolution can truly achieve therapeutic effects.

There will be other fronts to explore with the full set of CDNs. A large database will facilitate the detection of negative selection which has eluded detection (*Chen et al., 2022b*). Chen et al. have analyzed a curious phenomenon in somatic evolution, which they term 'quasi-neutral evolution' (*Chen et al., 2019*). It will also be possible to study the evolution of mutation mechanisms in cancer cells based on such large samples (*Jackson and Loeb, 1998*; *Ruan et al., 2020*). This last topic is addressed in **Appendix 1 Note 5** (*Appendix 1—figure 2*). Finally, at the center of evolutionary genetics is the multi-genic interactions that control complex phenotypes such as human diseases (e.g. diabetes; *Vujkovic et al., 2020*; *Lagou et al., 2023*; *Xue et al., 2023*; *Suzuki et al., 2024*), genetics of speciation (*Chen et al., 2022b*; *Pan et al., 2022b*) and the emergence of viral strains (*Deng et al., 2022*; *Pan et al., 2022a*). Cancers may be the first such complex genetic systems that can be unraveled thanks to the massively repeated evolution. As cancer genomics is increasingly adopted in cancer treatment, these benefits should become apparent when **n** reaches $10^5$ for most cancer types.

## Methods
### Data collection

Single nucleotide variation (SNV) data for the TCGA cohort was downloaded from the GDC Data Portal (https://portal.gdc.cancer.gov/, data version 2022-02-28). Only mutations identified by at least two pipelines were included in this study. Mutations were further filtered based on their population frequency recorded in the Genome Aggregation Database (*gnomAD*, version v2.1.1), with an upper threshold of 1‰. We focused on coding region mutations of missense, nonsense, and synonymous types on autosomes. The mutation load for each patient was defined as the sum of these three types of mutations. Patients with a mutation load exceeding 3000 were identified as having a mutator phenotype and were excluded from our analysis. In total, 7369 samples representing 12 cancer types were included for mutation analysis.

Additional mutation data was acquired from the AACR Project GENIE Consortium via cBioPortal. Due to the prevalence of targeted sequencing within the dataset, filtering was implemented to ensure the inclusion of samples with sequencing assays encompassing all exonic regions of target genes. Furthermore, sample-level filtering was performed to guarantee a unique sequencing sample per patient. Germline filtering was applied to the resulting point mutations, removing mutations with SNP frequencies exceeding 0.0005 in any subpopulation annotated by gnomAD. To exclude patients exhibiting hypermutator phenotypes, mutation loads were scaled to the whole exon level with $\tilde{n}_L = n_l \cdot \frac{L}{l}$, where $n_l$ and $l$ represent the mutation load and genomic length of target sequencing region, respectively. $L$ denotes the genome-wide whole coding region length. Patients with $\tilde{n}_L > 3000$ were subsequently excluded, consistent with the threshold employed for the TCGA dataset.

### Calculation of missense and synonymous site number ($L_A$ and $L_S$)

The idea of missense or synonymous sites originates from the question that how many missense or synonymous mutations would be expected if each site of the genome were mutated once given background mutation patterns. Here, the background mutation patterns refer to intrinsic biases in the mutational process, such as the over-representation of C>T (or G>A) mutations at CpG sites due to spontaneous deamination of 5-methylcytosine. In coding regions, fourfold degenerate sites are generally considered neutral, as any mutation path would not alter the encoded amino acid sequence. This analysis follows established methods at the single-base level to infer the expected number of missense and synonymous sites across the genome (*Gojobori et al., 1982*; *Wu and Maeda, 1987*; *Hartl and Clark, 1989*; *Martincorena et al., 2017*).

To illustrate the calculation process, we provide an example of synonymous site estimation. At four-fold degenerate sites throughout the genome, we tally the number of mutations from base $m$ to base $v$ as $n_{m>v}$ $(m, v \in \{A, C, G, T\}, m \neq v)$. The likelihood of observing a mutation from reference base $m$ to variant base $v$ will be $r_{m>v} = \frac{n_{m>v}}{N_m}$, where $N_m$ represents the number of fourfold degenerate site with reference base $m$. We then normalize all likelihoods by $R_{m>v} = \frac{r_{m>v}}{\sum r}$, where $\sum r$ represents the sum of likelihoods across 12 possible mutation paths, $R_{m>v}$ thus describes the relative probability of an occurred mutation to be $m > v$ at any site. For a given genomic coding region of length $L$, the synonymous site number will be:

$$L_S = \sum_L \delta_{m>v}^{syn} R_{m>v}$$

(S1)

with $\delta_{m>v}^{syn}$ being a Kronecker delta function where:
$\delta_{m>v}^{syn} = 1$ if is synonymous
$\delta_{m>v}^{syn} = 0$                    otherwise

Similarly, the expected number of missense sites $L_A$, is calculated as follows:

$$L_S = \sum_L \delta_{m>v}^{mis} R_{m>v}$$

(S2)

## Calculation of $A_i$ and $S_i$

For each mutated site, we track its number of recurrent mutations $i$ ($i>0$) across two mutation categories: missense and synonymous. Subsequently, we aggregate across entire coding region to count the number of sites harboring $i$ missense mutations ($A_i$) and synonymous mutations ($S_i$). For $i=0$, we define $A_0$ and $S_0$ as the estimated number of potential missense and synonymous sites that remain unmutated within the current sample size. $A_0 = L_A - \Sigma_{i>0} A_i, S_0 = L_s - \Sigma_{i>0} S_i$.

## AI-based mutation rate analysis

To capture the complex interplay of genomic and epigenomic factors influencing mutation susceptibility, we employed pre-trained artificial intelligence (AI) models from *Dig*, an aggregated tool combining deep learning and probabilistic models (*Sherman et al., 2022*). Downloaded from the *Dig* data portal (http://cb.csail.mit.edu/cb/DIG/downloads/), these models leverage a rich set of features encompassing both kilobase-scale epigenomic context (replicating timing, chromatin accessibility, etc.) and fine-grained base-pair level information (such as sequence context biases) to predict the site level mutation rate. For each cancer type, we re-fitted the pre-trained models with mutations analyzed in our study. The mutation rate for each site, scaled by population size, was obtained via the *element-Driver* function within Dig, and was represented by *EXP_SNV* from the final results. For a closer look at mutation rate landscape of *PAX3*, we re-fitted the AI-model with point mutations from large intestine cancer. The mutation rates were generated site-by-site for the coding regions of *PAX3*.

Given the scarcity of mutated sites with recurrence $i \geq 3$ (comprising only 0.15% of all mutated sites), a rigorous statistical approach was adopted to assess the significance of mutability differences between these high-hit groups and low-hit groups. We implemented the procedure as follows:

(1) Raw significance level: For each recurrence group $i$ (containing $A_i$ sites), a one-sided Kolmogorov-Smirnov (K-S) test was employed to calculate a raw significance level (denoted as $p_0$) against the low-hit group.

(2) Resampling for Significance Pool: we resample $A_i$ sites from the entire pool of mutated sites with missense mutations. The significance $p_j$ from one-sided K-S test is calculated against the low hit group. The resampling process was repeated 100,000 times, generating a distribution of resampled significance levels, denoted as $\{p_j, j = 1, 2, ..., 100,000\}$.

(3) Adjusted Significance Level: the raw significance $p_0$ was then compared to the resampled significance pool $\{p_j, j = 1, 2, ..., 100,000\}$. The proportion of $p_0 < p_j$ was then calculated as the adjusted significance level that accounted for potential sampling effects.

## Motif-based mutability

For a given nucleotide base, we extended the sequence to each side by 1, 2, and 3 base pairs, producing sets of 3-mer, 5-mer, and 7-mer motifs, respectively. We then pooled point mutations from

12 cancer types to create a comprehensive dataset. For 3-mer and 5-mer motifs, we utilized synonymous mutations as the reference for mutation rate calculations. For 7-mer motifs, the vast number of possible sequence combinations ($4^7$=16,384) posed a challenge, as synonymous mutations alone might not adequately cover all potential contexts. To address this, we employed all singleton mutations (1-hit mutations) from both missense and synonymous categories for 7-mer motif analysis. This decision was based on the assumption that singleton mutations are less affected by selective pressures, supported by the genome-wide observation that the ratio of missense to synonymous singletons ($A_1/S_1$) approximates the ratio of unmutated missense to synonymous sites ($A_0/S_0$).

The site number for 3-mer and 5-mer motifs of a given context $c$ was calculated as follows:

$$L_{c,S} = \sum_L \delta^{syn}_{c,m>v} \cdot R_{m>v}$$

Which is an extension of **Equation S1**, with $\delta^{syn}_{c,m>v}$ being a Kronecker delta function where:
$\delta^{syn}_{c,m>v} = 1$ if base change of $m$ to $v$ (**m > v**) is synonymous under sequence context $c$.
$\delta^{syn}_{c,m>v} = 0$ otherwise.

The mutation rate then could be expressed as:

$$\mu_c = \frac{n^{syn}_{c,m>v}}{L_{c,s}} \tag{S3}$$

for 7-mer motifs, the calculation is:

$$L_c = L_{c,S} + L_{c,A} = \sum_L \delta^{syn}_{c,m>v} \cdot R_{m>v} + \sum_L \delta^{mis}_{c,m>v} \cdot R_{m>v}$$

$$\mu_c = \frac{n^{syn}_{c,m>v} + n^{mis}_{c,m>v}}{L_c} \tag{S4}$$

Where for **Equations S3 and S4**, $n^{syn}_{c,m>v}$ and $n^{mis}_{c,m>v}$ represent the mutation numbers with **m>v** being synonymous and missense under sequence context $c$, respectively. The mutation rate is then scaled as the expected mutation number per $10^5$ corresponding sequence motifs for better presentation.

Significance for motif enrichment (**Figure 4B**) mirrored the AI analysis. For each $i{\geq}3$ site, we calculated raw K-S $p$-values against motif mutabilities (denoted as $p_0$). These were then compared to a resampled significance pool $\{p_j, j = 1, 2, ..., 100,000\}$, with the proportion of $p_0 \leq p_j$ employed as the final $p$-value, depicting enrichment significance for highly mutable motifs in recurrence group $i$ against low hits.

## Consensus length comparison

To explore potential sequence motifs associated with recurrent mutations ($i{\geq}3$), we employ a sliding window of 10 bp stride to extract the local context from reference genome (**Figure 4—figure supplement 1**). We examined diverse window sizes (21, 41, 61, 81, and 101 bp) to capture potential motifs of varying lengths and distances to the mutated site. Consensus length of local contexts was measured by Hamming distance in pairwise comparisons of aligned windows (with same stride) between mutated sites.

To prioritize sequence similarities likely driven by mutational mechanisms rather than functional constraints or gene structure, we restricted consensus comparisons to non-homologous genes. This approach effectively mitigated potential biases arising from homologous genes (e.g., *KRAS* and *NRAS*) or repeated domains within a single gene (e.g., *FBXW7*). The statistical significance of observed consensus lengths was assessed using the K-S test, which compared the empirical distribution of consensus lengths against a Poisson distribution, with mean of $\lambda$ set to one-quarter of the window size, which reflects the expected distribution under random scenarios.

## Mutational signature analysis

The mutation load of each patient could be further decomposed to several known mutational processes, which is represented by mutational signatures. In general, each mutational signature

embodies the relative mutabilities across distinct mutational contexts. Leveraging single base substitution (SBS) signatures from COSMIC (v3.3), we employed the *SignatureAnalyzer* tool to quantify the contribution of each signature to individual mutational loads (**Kim et al., 2016**). For composition analysis in *Figure 5B*, we focused on signatures contributing at least 2% to the total mutations within a given cancer type, given that there are 79 mutational signatures in use for deconvolution.

To assess signature contribution changes across recurrence cutoffs ($i$), we grouped patients with mutations of recurrence $i \geq i^*$ and scaled signature contributions to 1 to cancel out the population size effect. Pairwise K-S test between different $i$'s is employed to determine whether signature contributions are significantly different under each $i^*$.

## Outlier model

The purpose of the outlier model is to investigate if high-hit sites could be explained by a fraction ($p$) of highly mutable sites (with mutability $\alpha$-fold higher). By definition, we have:

$$S_i = (1-p) \cdot L_S \cdot [nE(u)]^i + p \cdot L_S \cdot [\alpha \cdot nE(u)]^i \tag{S5}$$

With **n** represents the population size and **E(u)** denotes the average mutation rate per site per patient.

We let $p$ range from $10^{-5}$ to $10^{-2}$ and $\alpha$ from 1.1 to 1,000. For each ($p$, $\alpha$) pair, we solve **Equation S5** based on observed **S₁** ($i$=1) to obtain **E(u)**. Then, we calculate expected **Sᵢ** with $i$=2, 3. The ($p$, $\alpha$) pairs are filtered by imposing constraints grounded in observed **S₂** and **S₃** values. Specifically, we retained only those pairs whose expected **S₂** and **S₃** values resided within the 95% quantile range of a Poisson distribution with $\lambda$ set to the observed values. This filtering process yielded biologically plausible ($p$, $\alpha$) pairs that were then used to derive **S₄** and **S₅**. Finally, we computed the mean and standard deviation for $p$, $\alpha$, **S₄**, and **S₅** across all filtered pairs to capture their central tendencies and variability.

## CDN analysis in GENIE

To circumvent potential biases in **E(u)** estimation stemming from the varying target gene coverage across sequencing panels within the GENIE dataset, we leveraged **E(u)** values derived from the corresponding cancer types in the TCGA dataset. Specifically, we focused on 1-hit synonymous mutations within coding regions, as these are generally considered to be the least influenced by selective pressures in coding regions. Based on **Equation 1** from the main text, we have:

$$S_1 = L_S \cdot nE(u)\, e^{-(n-1)E(u)} \tag{S6}$$

where $e^{(n-1)E(u)}$ comes from approximation of $[1 - E(u)]^{(n-1)}$ from binomial distribution, and **n** is the population size in TCGA. The calculation for the threshold $i^*$, based on **Equation 10** from the main text, is:

$$i^* = \frac{log\left(\dfrac{\varepsilon}{L_A}\right)}{log\left(\dfrac{n_e E(u)}{1 + n_e E(u)}\right)} \tag{S7}$$

The only difference in **Equation S7** is that we use **nₑ** to represent the number of patients sequenced for a target gene in GENIE, considering the overlapping between different assays in use. In essence, we will have for each gene a CDN threshold $i^*$.

The comparison of CDNs between TCGA and GENIE are restricted to genes sequenced by GENIE panels. For CDNs identified in GENIE using **Equation S7**, we investigated their hit information and CDN identity within TCGA dataset. The CDN flow proportion depicted in **Figure 7C–E** represents the ratio of sites identified as CDN in GENIE to $A_i^*$ ($i$=0, 1, 2) of TCGA. $A_i^*$ mirrors **Aᵢ** but specifically considers the coding region sequenced by the GENIE panel. For CDNs identified in TCGA, the flow ratio is just the proportion of sites being identified as CDNs in GENIE. Notably, for CDNs identified in GENIE but lacking mutations in TCGA ($i$=0), $A_0^*$ is obtained using **Equation S2** with **L** being the length of coding region sequenced in GENIE.

## Acknowledgements

The authors gratefully acknowledge the following for their support in the initiation of the Cancer Driving Nucleotide (CDN) project: the First Affiliated Hospital, the Seventh Affiliated Hospital of Sun Yat-sen University; Cancer Center of Clifford Hospital, Jinan University; Cancer Hospital Chinese Academy of Medical Sciences, Shenzhen Center; Guangdong Academy of Medical Sciences, Guangdong Provincial People's Hospital. We thank the Kunming Institute of Zoology for valuable discussions on the CDN concept. We are also grateful to Weiwei Zhai, Qianfei Wang, and Weini Huang for their insightful comments and suggestions. Finally, we acknowledge the American Association for Cancer Research (AACR) and The Cancer Genome Atlas (TCGA) project for providing invaluable datasets and resources that have significantly advanced our understanding of cancer biology and improved patient outcomes. This work was supported by the National Natural Science Foundation of China (32150006 to CIW and XL, 32293193, 32293190, 32200493, and 32370659) to CIW, 82341092 to HJ Wen, the National Key Research and Development Projects of the Ministry of Science and Technology of China (2021YFC2301300, 2021YFC0863400), Guangdong Key Research and Development Program (No. 2022 No. B1111030001), Guangdong Basic and Applied Basic Research Foundation (No. 2023A1515010016), Yunnan Revitalization Talent Support Program Top Team (202405AS350022 to CIW and XL) and Yunnan Revitalization Talent Support Program Yunling Scholar Project (XL).

## Additional information

### Funding

| Funder | Grant reference number | Author |
|---|---|---|
| National Natural Science Foundation of China | 32150006 | Xuemei Lu Chung-I Wu |
| Guangdong Key R&D Project of China | 2022B1111030001 | Hai-Jun Wen |
| National Natural Science Foundation of China | 32293193 | Chung-I Wu |
| National Natural Science Foundation of China | 32293190 | Chung-I Wu |
| Yunnan Revitalization Talent Support Program Top Team | 202405AS350022 | Xuemei Lu Chung-I Wu |
| National Natural Science Foundation of China | 82341092 | Hai-Jun Wen |
| National Natural Science Foundation of China | 32200493 | Chung-I Wu |
| National Key Research and Development Program of China | 2021YFC2301300 | Chung-I Wu |
| National Key Research and Development Program of China | 2021YFC0863400 | Chung-I Wu |
| Yunnan Revitalization Talent Support Program Yunling Scholar Project | | Xuemei Lu |
| National Natural Science Foundation of China | 32370659 | Chung-I Wu |
| Guangdong Basic and Applied Basic Research Foundation | 2023A1515010016 | Chung-I Wu |

| Funder | Grant reference number | Author |
| --- | --- | --- |

The funders had no role in study design, data collection and interpretation, or the decision to submit the work for publication.

## Author contributions

Lingjie Zhang, Conceptualization, Data curation, Formal analysis, Visualization, Methodology, Writing – original draft, Project administration; Tong Deng, Data curation, Validation, Investigation; Zhongqi Liufu, Bingjie Chen, Validation, Investigation; Xueyu Liu, Validation, Visualization; Zheng Hu, Supervision, Validation, Investigation, Project administration; Chenli Liu, Validation, Project administration; Miles E Tracy, Writing – review and editing; Xuemei Lu, Conceptualization, Supervision, Validation, Investigation, Project administration; Hai-Jun Wen, Validation; Chung-I Wu, Conceptualization, Supervision, Funding acquisition, Validation, Investigation, Methodology, Project administration, Writing – review and editing

## Author ORCIDs

Lingjie Zhang  https://orcid.org/0000-0002-6506-4457
Zheng Hu  https://orcid.org/0000-0003-1552-0060
Xuemei Lu  https://orcid.org/0000-0001-6044-6002
Hai-Jun Wen  https://orcid.org/0000-0001-8676-1254
Chung-I Wu  https://orcid.org/0000-0001-7263-4238

Reviewer #1 (Public review): https://doi.org/10.7554/eLife.99340.3.sa1
Reviewer #2 (Public review): https://doi.org/10.7554/eLife.99340.3.sa2
Author response https://doi.org/10.7554/eLife.99340.3.sa3

## Additional files

### Supplementary files

• Supplementary file 1. All CDN sites with population allele frequency annotation. '**CDN_sites.i ≥ 20**' presents CDN sites with total hits ≥20 in each cancer type. '**CDN.Missense.thres_3**' provides all CDN sites analyzed in this study, ranked in decreasing order based on the highest recurrence across 12 cancer types. '**CDN.Missense.gnomAD**' presents the *gnomAD* population allele frequency of all missense CDNs. '**Synonymous_high_hits**' lists the synonymous mutations potentially under selection, while '**Synonymous_high_hits.gnomAD**' provides their corresponding allele frequency annotations from *gnomAD*.

• MDAR checklist

### Data availability

The key scripts used in this study are available at GitLab, copy archived at *Zhang, 2024*. A subset of key example files for breast cancer analysis can be found in the "/example_data_files" directory. The complete list of CDNs analyzed in this study is provided in *Supplementary file 1*.

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

# Appendix 1

## 1. Literature support for CDNs identified in breast cancer

Verification of site level positive selection in cancer genome has primarily focused on canonical cancer drivers. For non-canonical candidates with experimentally proven tumorigenic activity, CDN sites within these genes emerge as potential key drivers due to their statistically stronger selective advantage. Under this premise, we search for literature evidence for genes harboring CDN sites in breast cancer.

Among the 17 genes with CDN sites in breast cancer, 11 are recognized as canonical drivers by all three major driver gene lists (*Appendix 1—table 1*). 4 genes (*CDC42BPA*, *ERBB3*, *KIF1B*, *NUP93*), despite lacking inclusion in canonical breast cancer driver lists, possess explicit experimental support indicating their driving roles in breast tumorigenesis. *HIST1H3B* has been recognized as a driver gene in breast cancer by IntOGen, corroborated by literatures supporting its association with breast cancer. The four mutation recurrences with R6C alteration in amino acid sequence in *RARS2* have been proposed to be linked to defects in mitochondrial transport, the explicit role of *RARS2* in breast cancer tumorigenesis remains to be explored.

**Appendix 1—table 1.** literature support for CDN genes in breast cancer.

| Gene Id | Gene Name | Support |
|---|---|---|
| *AKT1* | v-akt murine thymoma viral oncogene homolog 1 | ① ② ③ |
| *CDC42BPA* | CDC42 binding protein kinase alpha (DMPK-like) | *Unbekandt and Olson, 2014*; *Collins et al., 2018*; *Kwa et al., 2021*; *Jiang et al., 2023* |
| *CDH1* | cadherin 1, type 1, E-cadherin (epithelial) | ① ② ③ |
| *ERBB2* | v-erb-b2 avian erythroblastic leukemia viral oncogene homolog 2 | ① ② ③ |
| *ERBB3* | v-erb-b2 avian erythroblastic leukemia viral oncogene homolog 3 | *Holbro et al., 2003*; *Xue et al., 2006*; *Hamburger, 2008*; *Sithanandam and Anderson, 2008*; *Stern, 2008*; *Huang et al., 2010* |
| *FGFR2* | fibroblast growth factor receptor 2 | ① ② ③ |
| *FOXA1* | forkhead box A1 | ① ② ③ |
| *GATA3* | GATA binding protein 3 | ① ② ③ |
| *HIST1H3B* | histone cluster 1, H3b | ① ② ③ *Xie et al., 2019*; *Wang et al., 2023*\* |
| *KIF1B* | kinesin family member 1B | *Munirajan et al., 2008*; *Yu and Feng, 2010*; *Liu et al., 2022* |
| *KRAS* | Kirsten rat sarcoma viral oncogene homolog | ① ② ③ |
| *NUP93* | nucleoporin 93 kDa | *Bersini et al., 2020*; *Nataraj et al., 2022* |
| *PIK3CA* | phosphatidylinositol-4,5-bisphosphate 3-kinase, catalytic subunit alpha | ① ② ③ |
| *PTEN* | phosphatase and tensin homolog | ① ② ③ |
| *RARS2* | arginyl-tRNA synthetase 2, mitochondrial | *Wang et al., 2020*\* |
| *SF3B1* | splicing factor 3b, subunit 1, 155 kDa | ① ② ③ |
| *TP53* | tumor protein p53 | ① ② ③ |

The serial number corresponds to the inclusion of target gene in the following driver gene list: ① CGC Tier-1 list, ② IntOGen, ③ Bailey's list.
The inclusion necessitates that the target gene is annotated as a cancer driver in breast cancer.
\*ambiguous, meaning the literature indicates an association between the candidate gene and breast cancer, but lacks explicit experimental evidence.

## 2. The impact of k for gamma-binomial model

From *Equation 2* in the main text, $S_i$ is affected by two terms: $G$ and **nE(u)**, where $G$ is:

$$G = L_s \cdot g(i, k) = L_s \cdot C_{k+i-1}^i \cdot \frac{1}{k^i}$$

$L_s$ is a constant value given specific cancer type, here we demonstrate how $\varphi(i,k)$ varies with respect to $i$ and $k$, and elucidate why $S_i$ of $k=1$ indicates the upper bound of CDN cutoff.

Considering the fold change of $G$ from $i$-1 to $i$.

$$\varphi(i,k) = \frac{g(i,k)}{g(i-1,k)} = \frac{C^i_{k+i-1} \cdot \frac{1}{k^i}}{C^i_{k+i-2} \cdot \frac{1}{k^{i-1}}}$$

(S8)

*Appendix 1—figure 1* illustrates how $\varphi(i,k)$ changes with $i$ and $k$. With $k$ range from 0.1 to 10, the curve of $\varphi(i,k)$ elucidates the extent to which $G$ would impact $S_i$ with each increment of $i$. As detailed in **Section 4**, $k$ will be >1 for biological significance. In such cases, $\varphi(i,k)$ will always be < 1, meaning $G$ will synergistically collaborate with **nE(u)** to decrease $S_i$. The diminishing impact of $\varphi(i,k)$ intensifies as $k$ increases. A higher $k$ value would suggest that, for most sites across the genome, mutability falls within a narrow range of >0. In an extreme case, when $k=10$, $Pois(i \vee \lambda) = 0.145$ at $i=20$, which is 2 orders weaker than **nE(u)** for cancer types in TCGA. In practice, $k$ is usually estimated to be between 2–5, depending on the cancer types being investigated (*Appendix 1—table 2*). Consequently, the reduction of $S_i$ with each increase of $i$ is predominantly governed by **nE(u)**, and the $S_i$ values with $k=1$ represent the upper limit driven solely by mutational force.

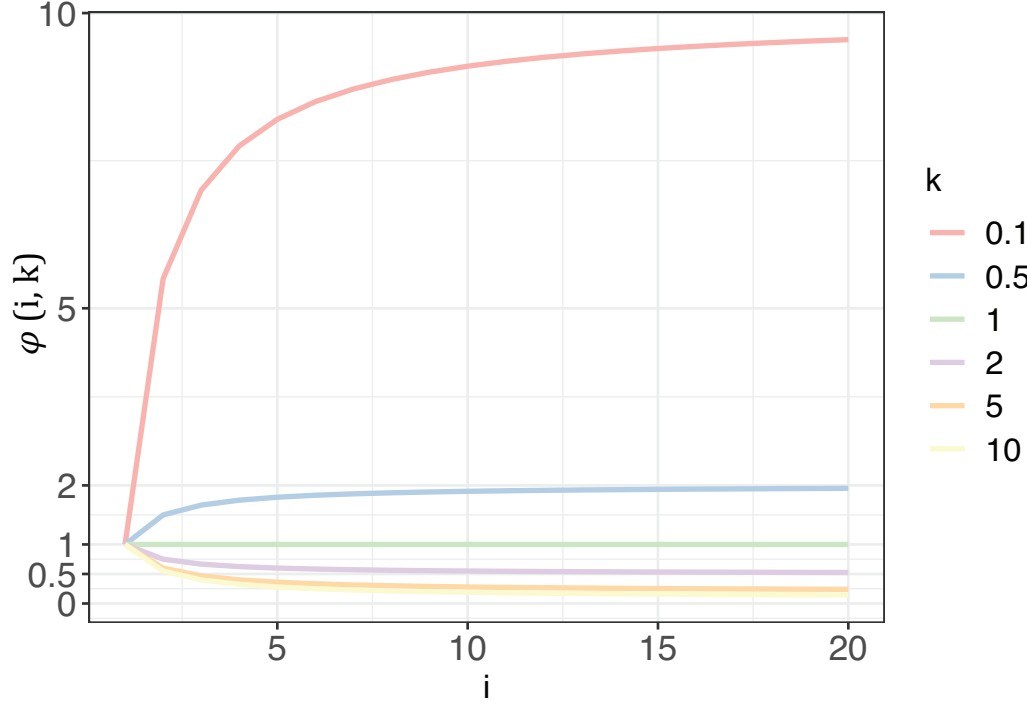

**Appendix 1—figure 1.** The trend of $\lambda$ with each increase of recurrence ($i$, the x-axis) under different shape parameters of the gamma distribution ($k$, designated by different colors).

**Appendix 1—table 2.** $k$ estimated from 12 cancer types.

| Cancer type | $k$ |
| --- | --- |
| Breast | 5.05 |
| CNS | 2.59 |
| Endometrium | 5.49 |
| Kidney | 7.70 |

*Appendix 1—table 2 Continued on next page*

*Appendix 1—table 2 Continued*

| Cancer type | k |
|---|---|
| Large intestine | 4.76 |
| Liver | 5.23 |
| Lung | 2.62 |
| Ovary | 4.30 |
| Prostate | 3.60 |
| Stomach | 4.17 |
| Upper-AD tract | 4.14 |
| Urinary tract | 6.14 |
| merged set[*] | 3.27 |

Note:- Estimation of *k* is derived from negative binomial regression, based on synonymous changes aggregated by the 3 bp local context at mutated sites across all coding genes. The estimation method is implemented in package dndscv.
[*]The merged set contains mutation information from all 12 cancer types.

## 3. The impact of negative selection on shape parameter *k*

Across various studies aiming to depict mutability variation across genome under the gamma distribution, the shape parameter *k* is always pivotal. Adopting a dichotomous perspective, we inquire into how *k* compares to 1 under large sample sizes ($n \geq 10^6$). This inquiry is fundamentally linked to the prevalence of negative selection across the cancer genome, as the observed mutation abundance is an amalgamation of mutational and selection forces. In scenarios where purifying selection extensively operates throughout the genome in cancer evolution, even for synonymous sites (*Sharp and Li, 1987*; *Plotkin and Kudla, 2011*; *Gartner et al., 2013*; *Chu and Wei, 2019*), most genomic sites would not exhibit mutations, resulting in *k* being ≤1. In attempts to detect negative selection signals in cancer, researchers typically identify only a limited number of genes (*Luo et al., 2008*; *Van den Eynden et al., 2016*; *Zapata et al., 2018*; *Bányai et al., 2021*). In a CRISPR-Cas9 loss-of-function screen covering 16,540 genes conducted across 558 cancer cell lines, only approximately 6% of genes are under strong negative selection in at least 90% of the cell lines (*De Kegel and Ryan, 2019*). In an in-house mutation accumulation experiment carried out in HCT116 (a human colorectal carcinoma cell line, data not published), the proportion of mutations under strong negative selection is 0.66% with a selection coefficient (*s*) of –0.6 (indicating that the survivability of the mutant is 40% of the wildtype). These evidences, in concordance with quasi-neutrality of cancer evolution, suggest that purifying selection is indeed rare in cancer evolution. The mutability for the majority of genomic sites is greater than 0, with shape parameter *k*>1.

## 4. Detailed derivation for negative binomial distribution and approximation of $i^*/n$

The derivation of *Equation 5* from joint distribution of Gamma-Poisson distribution is well presented in statistically analysis. Here, we assume that the mutation recurrence (*i*) observed at site level across the genome follows a Poisson distribution of $Pois(i|\lambda)$, where the expected number of mutation recurrence $\lambda$ follows a Gamma distribution of $Gamma(\lambda|k, \theta)$, with *k* and $\theta$ being the shape and scale parameters, respectively. Then, the joint probability density function for *i* can be expressed as:

$$f(i|k, \theta) = \int Pois(i|\lambda) \cdot Gamma(\lambda|k, \theta) \cdot d\lambda$$

$$= \int \frac{\lambda^i e^{-\lambda}}{i!} \cdot \frac{1}{\Gamma(k)\theta^k} \lambda^{k-1} e^{\frac{-\lambda}{\theta}} \cdot d\lambda$$

$$= \frac{\theta^{-k}}{i!\Gamma(k)} \int \lambda^i e^{\lambda} \cdot \lambda^{k-1} e^{\frac{\lambda}{\theta}} \cdot d\lambda$$

$$= \frac{\theta^{-k}}{i!\Gamma(k)} \int \lambda^{(i+k)-1} e^{-\left(1+\frac{1}{\theta}\right)\lambda} \cdot d\lambda \tag{S9}$$

Now, we make use of the probability density function of Gamma distribution,

$$\int Gamma(\lambda|k,\theta) = 1$$

Which is:

$$\int \frac{1}{\Gamma(k)\theta^k} \lambda^{k-1} e^{-\frac{\lambda}{\theta}} \cdot d\lambda = 1$$

Therefore,

$$\int \lambda^{k-1} e^{\frac{-\lambda}{\theta}} \cdot d\lambda = \Gamma(k)\left(\frac{1}{\theta}\right)^{-k} \tag{S10}$$

Comparing with *Equation S10*, *Equation S9* can be rewrite as:

$$f(i|k,\theta) = \frac{\theta^{-k}}{i!\Gamma(k)} \cdot \left(1+\frac{1}{\theta}\right)^{-(i+k)} \Gamma(i+k)$$

$$\frac{\theta^{-k}}{i!\Gamma(k)} \cdot \left(\frac{\theta}{1+\theta}\right)^{i+k} \Gamma(i+k)$$

$$= \frac{\Gamma(i+k)}{\Gamma(i+1)\Gamma(k)} \left(\frac{1}{1+\theta}\right)^k \left(\frac{\theta}{1+\theta}\right)^i \tag{S11}$$

Note that the mean for Gamma distribution is $k\theta = nE(u)$, which leads to $\theta = \frac{nE(u)}{k}$. Then, the negative-binomial form of *Equation S11* could be further expressed as:

$$f(i|k,\theta) = \frac{\Gamma(i+k)}{\Gamma(i+1)\Gamma(k)} \left(\frac{k}{k+nE(u)}\right)^k \left(\frac{nE(u)}{k+nE(u)}\right)^i$$

$$\frac{\Gamma(i+k)}{\Gamma(i+1)\Gamma(k)} k^k \left[nE(u)\right]^i \left[k+nE(u)\right]^{-k-i}$$

Which is *Equation 6* from the main text.

With $k=1$, $nE(u) = k\theta = \theta$, *Equation S11* then transforms to:

$$f(i|1,\theta) = \frac{\Gamma(i+1)}{\Gamma(i+1)\Gamma(1)} \left(\frac{1}{1+\theta}\right)^1 \left(\frac{\theta}{1+\theta}\right)^i$$

$$\left(\frac{1}{1+\theta}\right) \cdot \left(1-\frac{1}{1+\theta}\right)^i$$

$$\left(\frac{1}{1+nE(u)}\right) \cdot \left(1-\frac{1}{1+nE(u)}\right)^i$$

Which is a geometric distribution with $p = \frac{1}{1+nE(u)}$.

For the approximation of $i^*/$**n**, we let $\varepsilon=1$, then *Equation 10* from main text could be rewritten as:

$$i^* \cdot \log\left(\frac{1}{1+\frac{1}{nE(u)}}\right) = \log\left(\frac{1}{L_A}\right)$$

$$i^* \cdot \log\left(1 + \frac{1}{nE\left(u\right)}\right) = \log\left(L_A\right) \tag{S12}$$

For the left side of *Equation S12*, we use the first-order Tayler expansion,

$$i^* \cdot \log\left(1 + \frac{1}{nE(u)}\right) = \log(L_A)$$

Substitute this to *Equation S12*, we have:

$$\frac{i^*}{n} = \log(L_A) \cdot E(u)$$

Which is *Equation 11* from main text.

## 5. Probing mutation rate variation with large samples

With large sample size sequenced, the data will yield an additional benefit by revealing the evolution of the mutation rate itself. Given that the mutation rate per site is extremely small, the evolution of mutation rate itself has been a most challenging issue (*André and Godelle, 2006*; *Lynch, 2010*; *Lynch, 2011*; *Lynch et al., 2016*; *Ruan et al., 2020*; *Wei et al., 2022*). In particular, without the check of selection, the mutation rate is liable to be trapped in the runaway evolution (*Ruan et al., 2020*).

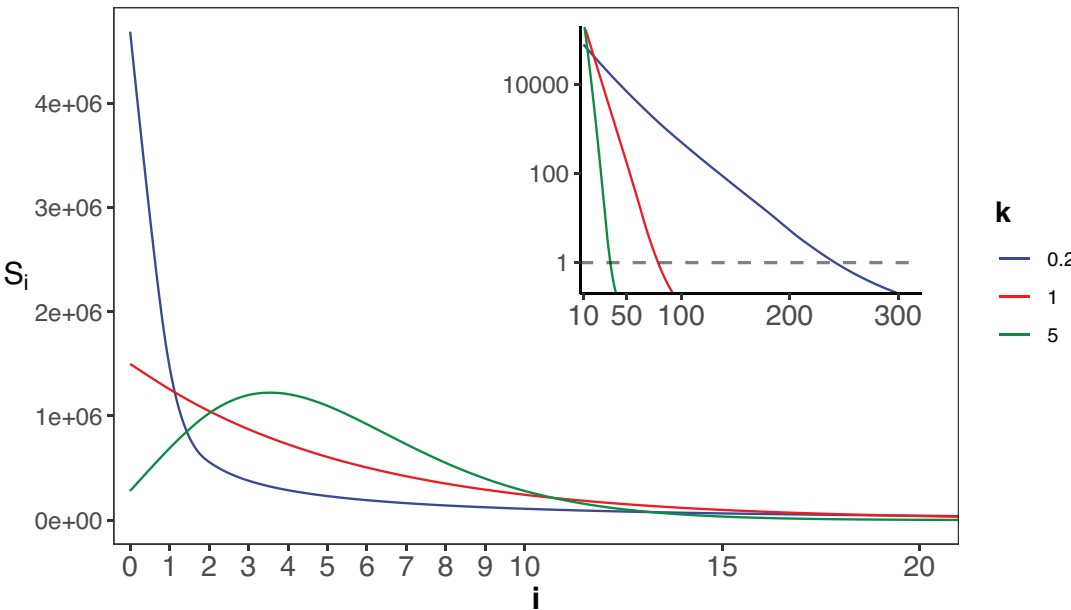

**Appendix 1—figure 2.** The gamma distribution of recurrences (i) under different shapes. With $E(u)=5 \times 10^{-6}$, we set the shape parameter *k* to 0.2, 1 and 5, represented by three distinct colors. The site number of synonymous recurrence *i* ($S_i$) is indicated on Y-axis. In the context of a large sample size ($n=10^6$), the $S_i$ distribution clearly distinguishes between different *k* values, mitigating the overdispersion issue encountered in smaller sample sizes. The inset depicts the distribution on a log10 scale for $i \geq 10$, with a horizontal dashed line indicating $S_i=1$, where $i^*$ is the CDN cutoff.

The theory of mutation rate of evolution should be based on the distribution of the per-site mutation rate across the genome. However, the empirical data so far only yield the mean. In particular, the spectrum of $S_i$'s for *i*'s close to 1 would be most informative about the evolution of the mutation mechanism. *Appendix 1—figure 2* shows the $S_i$ spectrum with *k*=0.2, 1 or 5 in a Gamma distribution. Note the mode of the distribution (i.e. the peak of the curve) among the 3 curves, which is at 0 or >0 depending on whether *k*≤1 or>1. Clearly, the observed $S_i$'s can distinguish among the three distributions only when **n** is very large. The implications of such distributions for the theory of mutation rate evolution are addressed in Discussion.

