## [Editor Report · eLife Assessment]

This **important** paper introduces a theoretical framework and methodology for identifying Cancer Driving Nucleotides (CDNs), primarily based on single nucleotide variant (SNV) frequencies. A variety of **solid** approaches indicate that a mutation recurring three or more times is more likely to reflect selection rather than being the consequence of a mutation hotspot. The method is rigorously quantitative, though the requirement for larger datasets to fully identify all CDNs remains a noted limitation. The work will be of broad interest to cancer geneticists and evolutionary biologists.

---

## [Referee Report · Reviewer #1 (Public review)]

The authors developed a rigorous methodology for identifying all Cancer Driving Nucleotides (CDNs) by leveraging the concept of massively repeated evolution in cancer. By focusing on mutations that recur frequently in pan-cancer, they aimed to differentiate between true driver mutations and neutral mutations, ultimately enhancing the understanding of the mutational landscape that drives tumorigenesis. Their goal was to call a comprehensive catalogue of CDNs to inform more effective targeted therapies and address issues such as drug resistance.

Strengths

(1) The authors introduced a concept of using massively repeated evolution to identify CDNs. This approach recognizes that advantageous mutations recur frequently (at least 3 times) across cancer patients, providing a lens to identify true cancer drivers.

(2) The theory showed the feasibility of identifying almost all CDNs if the number of sequenced patients increases to 100,000 for each cancer type.

Weaknesses

(1) No novel true driver mutations were identified in this study.

(2) Different cancer types have unique mutational landscapes. The methodology, while robust, might face challenges in uniformly identifying CDNs across various cancers with distinct genetic and epigenetic contexts.

(3) The statement "In other words, the sequences surrounding the high-recurrence sites appear rather random.". Since it was a pan-cancer analysis, the unique patterns of each cancer type could be strongly diluted in the pan-cancer data.

---

## [Referee Report · Reviewer #2 (Public review)]

Summary:

The authors propose that cancer driver mutations can be identified by Cancer Driving Nucleotides (CDNs). CDNs are defined as SNVs that occur frequently in genes. There are many ways to define cancer driver mutations, and strengths and weaknesses are the reliance of statistics to define them.

Strengths:

There are many well-known approaches and studies that have already identified many canonical driver mutations. A potential strength is that mutation frequencies may be able to identify, as yet, unrecognized driver mutations. They use of a previously developed method to estimate mutation hotspots across the genome (Dig, Sherman et al 2022). This publication has already used cancer sequence data to infer driver mutations based on higher than expected mutation frequencies. The advance here is to further illustrate that recurrent mutations (estimated at 3 or more mutations (CDNs) at the same base) are more likely to be the result of selection for a driver mutation (Fig 3). Further analysis indicates that mutation sequence context (Fig 4) or mutation mechanisms (Fig 5) are unlikely to be major causes for recurrent point mutations. Finally, they calculate (Fig 6) that most driver mutations identifiable by the CDN approach could be identified with about 100,000 to one million tumor coding genomes.

Weaknesses:

The manuscript does provide specific examples where recurrent mutations identify known driver mutations, but does not identify "new" candidate driver mutations. Driver mutation validation is difficult and at least clinically, frequency (ie observed in multiple other cancer samples) is indeed commonly used to judge if a SNV has driver potential. The method would miss alternative ways to trigger driver alterations (translocations, indels, epigenetic, CNVs). Nevertheless, the value of the manuscript is its quantitative analysis of why mutation frequencies can identify cancer driver mutations.

---

## [Author Response]

The following is the authors’ response to the original reviews.

**eLife assessment** This valuable paper reports a theoretical framework and methodology for identifying Cancer Driving Nucleotides (CDNs), primarily based on single nucleotide variant (SNV) frequencies. A variety of solid approaches indicate that a mutation recurring three or more times is more likely to reflect selection rather than being the consequence of a mutation hotspot. The method is rigorously quantitative, though the requirement for larger datasets to fully identify all CDNs remains a noted limitation. The work will be of broad interest to cancer geneticists and evolutionary biologists.

The key criticism “the requirement for larger datasets to fully identify all CDNs remains a noted limitation” that is also found in both reviews. We have clarified the issue in the main text, the relevant parts, from which are copied below. The response below also addresses many comments in the reviews. In addition, Discussion of eLife-RP-RA-2024-99341 has been substantially expanded to answer the questions of Reviewer 2.

We shall answer the boldface comment in three ways. First, it can be answered using GENIE data. Fig. 7 of the main text (eLife-RP-RA-2024-99340) shows that, when n increases from ~ 1000 to ~ 9,000, the numbers of discovered CDNs increase by 3 – 5 fold, most of which come from the two-hit class. Hence, the power of discovering more CDNs with larger datasets is evident. By extrapolation, a sample size of 100,000 should be able to yield 90% of all CDNs, as calculated here. (Fig. 7 also addresses the queries of whether we have used datasets other than TCGA. We indeed have used all public data, including GENIE and COSMIC.)

Second, the power of discovering more cancer driver genes by our theory is evident even without using larger datasets. Table 3 of the companion study (eLife-RP-RA-2024-99341) shows that, averaged across cancer types, the conventional method would identify 45 CDGs while the CDN method tallies 258 CDGs. The power of the CDN method is demonstrated. This is because the conventional approach has to identify CDGs (cancer driver genes) in order to identify the CDNs they carry. However, many CDNs occur in non-CDGs and are thus missed by the conventional approach. In Supplementary File S2, we have included a full list of CDNs discovered in our study, along with population allele frequency annotations from *gnomAD*. The distribution patterns of these CDNs across different cancer types show their pan-cancer properties as further explored in the companion paper.

Third, while many, or even most CDNs occur in non-CDGs and are thus missed, the conventional approach also includes non-CDN mutations in CDGs. This is illustrated in Fig. 5 of the companion study (eLife-RP-RA-2024-99341) that shows the adverse effect of misidentifications of CDNs by the conventional approach. In that analysis, the gene-targeting therapy is effective if the patient has the CDN mutations on *EGFR*, but the effect is reversed if the *EGFR* mutations are non-CDN mutations.

**Reviewer #1 (Public Review):**
The authors developed a rigorous methodology for identifying all Cancer Driving Nucleotides (CDNs) by leveraging the concept of massively repeated evolution in cancer. By focusing on mutations that recur frequently in pan-cancer, they aimed to differentiate between true driver mutations and neutral mutations, ultimately enhancing the understanding of the mutational landscape that drives tumorigenesis. Their goal was to call a comprehensive catalogue of CDNs to inform more effective targeted therapies and address issues such as drug resistance.Strengths(1) The authors introduced a concept of using massively repeated evolution to identify CDNs. This approach recognizes that advantageous mutations recur frequently (at least 3 times) across cancer patients, providing a lens to identify true cancer drivers.(2) The theory showed the feasibility of identifying almost all CDNs if the number of sequenced patients increases to 100,000 for each cancer type.Weaknesses(1) The methodology remains theoretical and no novel true driver mutations were identified in this study.

We now address the weakness criticism, which is gratefully received.

The second part of the criticism (no novel true driver mutations were identified in this study) has been answered in the long responses to eLife assessment above. The first part “*The methodology remains theoretical*” is somewhat unclear. It might be the lead to the second part. However, just in case, we interpret the word “theoretical” to mean “the lack of experimental proof” and answer below.

As Reviewer #1 noted, a common limitation of theoretical and statistical analyses of cancer drivers is the need to validate their selective advantage through in vitro or in vivo functional testing. This concern is echoed by both reviewers in the companion paper (eLife-RP-RA-2024-99341), prompting us to consider the methodology for functional testing of potential cancer drivers. An intuitive approach would involve introducing putative driver mutations into normal cells and observing phenotypic transformation in vitro and in vivo. In a recent stepwise-edited human melanoma model, Hodis et al. demonstrated that disease-relevant phenotypes depend on the “correct” combinations of multiple driver mutations (Hodis et al. 2022). Other high-throughput strategies can be broadly categorized into two approaches: (1) introducing candidate driver mutations into pre-malignant model systems that already harbor a canonical mutant driver (Drost and Clevers 2018; Grzeskowiak et al. 2018; Michels et al. 2020) and (2) introducing candidate driver mutations into growth factor-dependent cell models and assessing their impact on resulting fitness (Bailey et al. 2018; Ng et al. 2018). The underlying assumption of these strategies is that the fitness outcomes of candidate driver mutations are influenced by pre-existing driver mutations and the specific pathways or cancer hallmarks being investigated. This confines the functional test of potential cancer driver mutations to conventional cancer pathways. A comprehensive identification of CDNs is therefore crucial to overcome these limitations. In conjunction with other driver signal detection methods, our study aims to provide a more comprehensive profile of driver mutations, thereby enabling the functional testing of drivers involved in non-conventional cancer evolution pathways.

(2) Different cancer types have unique mutational landscapes. The methodology, while robust, might face challenges in uniformly identifying CDNs across various cancers with distinct genetic and epigenetic contexts.

We appreciate the comment. Indeed, different cancer types should have different genetic and epigenetic landscapes. In that case, one may have expected CDNs to be poorly shared among cancer types. However, as reported in Fig. 4 of the companion study, the sharing of CDNs across cancer types is far more common than the sharing of CDGs (Cancer Driving Genes). We suggest that CDNs have a much higher resolution than CDGs, whereby the signals are diluted by non-driver mutations. In other words, despite that the mutational landscape may be cancer-type specific, the pan-cancer selective pressure may be sufficiently high to permit the detection of CDN sharing among cancer types.

Below, we shall respond in greater details. Epigenetic factors, such as chromatin states, methylation/acetylation levels, and replication timing, can provide valuable insights when analyzing mutational landscapes at a regional scale (Stamatoyannopoulos et al. 2009; Lawrence et al. 2013; Makova and Hardison 2015; Baylin and Jones 2016; Alexandrov et al. 2020; Abascal et al. 2021; Sherman et al. 2022). However, at the site-specific level, the effectiveness of these covariates in predicting mutational landscapes depends on their integration into a detailed model. Overemphasizing these covariates could lead to false negatives for known driver mutations (Hess et al. 2019; Elliott and Larsson 2021). In figure 3B of the main text, we illustrate the discrepancy between the mutation rate predictions from *Dig* and empirical observation. Ideally, no covariates would be needed under extensive sample sizes, where each mutable genomic sites would have sufficient mutations to yield a statistic significance and consequently, synonymous mutations would be sufficient for the characterization of mutational landscape. In this sense, the integration of mutational covariates represents a compromise under current sample size. In our study, the effect of unique mutational landscapes is captured by E(u), the mean mutation rate for each cancer type. We further accounted for the variability of site-level mutability using a gamma distribution. The primary goal of our study is to determine the upper limit of mutation recurrences under mutational mechanisms only. While selection force acts blindly to genomic features, mutational hotspots should exhibit common characteristics determined by their underlying mechanisms. In the main text, we attempted to identify such shared features among CDNs. Until these mutational mechanisms are fully understood, CDNs should be considered as potential driver mutations.

(3) L223, the statement "In other words, the sequences surrounding the high-recurrence sites appear rather random.". Since it was a pan-cancer analysis, the unique patterns of each cancer type could be strongly diluted in the pan-cancer data.

We now state that the analyses of mutation characteristic have been applied to the individual cancer types and did not find any pattern that deviates from randomness. Nevertheless, it may be argued that, with the exception of those with sufficiently large sample sizes such as lung and breast cancers, most datasets do not have the power to reject the null hypothesis. To alleviate this concern, we applied the ResNet and LSTM/GRU methods for the discovery of potential mutation motifs within each cancer type. All methods are more powerful than the one used but the results are the same – no cancer type yields a mutation pattern that can reject the null hypothesis of randomness (see below).

As a positive control, we used these methods for the discovery of splicing sites of human exons. When aligned up with splicing site situated in the center (position 51 in the following plot), the sequence motif would look like:

**Author response image 1. sa3fig1:** 5-prime.

**Author response image 2. sa3fig2:** 3-prime.

However, To account for the potential influence of distance from the mutant site in motif analysis, we randomly shuffled the splicing sites within a specified window around the alignment center, and their sequence logo now looks like:

**Author response image 3. sa3fig3:** 5-prime shuffled.

**Author response image 4. sa3fig4:** 3-prime shuffled.

**Author response image 5. sa3fig5:** random sequences from coding regions.

The classification results of the shuffled 5-prime (donner), 3-prime (acceptor) and random sequences from coding regions (Random CDS) are presented in the Author response table 1 (The accuracy for the aligned results, which is approximately 99%, is not shown here).

**Author response table 1. sa3table1:** 

				TRAIN accuracy						TEST accuracy
						Random			random	
Model	# layers	# parameters	ALL	donner	acceptor	CDS	ALL	donner	acceptor	CDS
resNet1	46	3,721	0.77	0.77	0.76	0.79	0.71	0.72	0.69	0.72
resNet2	165	64,445	0.96	0.96	0.95	0.95	0.76	0.77	0.77	0.73
deepGRU	23	144,925	0.86	0.85	0.84	0.89	0.79	0.79	0.76	0.8
deepLSTM	6	24,445	0.85	0.82	0.87	0.86	0.79	0.76	0.82	0.78

With the positive results from these positive controls (splicing site motifs) validating our methodology, we applied the same model structure to the train and test of potential mutational motifs of CDN sites. All models achieved approximately 50% accuracy in CDN motif analysis, suggesting that the sequence contexts surrounding CDN sites are not significantly different from other coding regions of the genome. This further implies that the recurrence of mutations at CDN sites is more likely driven by selection rather than mutational mechanisms.

Note that this preliminary analysis may be limited by insufficient training data for CDN sites. Future studies will require larger sample sizes and more sophisticated models to address these limitations.

(4) To solidify the findings, the results need to be replicated in an independent dataset.

Figure 7 validates our CDN findings using the GENIE dataset, which primarily consists of targeted sequencing data from various panels. By focusing on the same genomic regions sequenced by GENIE, we observed a 3-5 fold increase in the number of discovered CDNs as sample size increased from approximately 1000 to 9000. Moreover, the majority of CDNs identified in TCGA were confirmed as CDNs in GENIE.

(5) The key scripts and the list of key results (i.e., CDN sites with i{greater than or equal to}3) need to be shared to enable replication, validation, and further research. So far, only CDN sites with i{greater than or equal to}20 have been shared.

We have now updated the “Data Availability” section in the main text, the corresponding scripts for key results are available on Gitlab at: https://gitlab.com/ultramicroevo/cdn_v1.

(6) The versions of data used in this study are not clearly detailed, such as the specific version of gnomAD and the version and date of TCGA data downloaded from the GDC Data Portal.

The versions of data sources have now been updated in the revised manuscript.

**Recommendations For The Authors:**
(1) L119, states "22.7 million nonsynonymous sites," but Table 1 lists the number as 22,540,623 (22.5 million). This discrepancy needs to be addressed for consistency.(2) Figure 2B, there is an unexplained drop in the line at i = 6 and 7 (from 83 to 45). Clarification is needed on why this drop occurs.(3) Figure 3A, for the CNS type, data for recurrence at 8 and 9 are missing. An explanation should be provided for this absence.(4) L201, the title refers to "100-mers," but L218 mentions "101-mers." This inconsistency needs to be corrected to ensure clarity and accuracy.(5) Figures 6 and 7 currently lack titles. Titles should be added to these figures to improve readability.

Thanks. All corrections have been incorporated into the revised manuscript.

**Reviewer #2 (Public Review):**
Summary:The authors propose that cancer-driver mutations can be identified by Cancer Driving Nucleotides (CDNs). CDNs are defined as SNVs that occur frequently in genes. There are many ways to define cancer driver mutations, and the strengths and weaknesses are the reliance on statistics to define them.Strengths:There are many well-known approaches and studies that have already identified many canonical driver mutations. A potential strength is that mutation frequencies may be able to identify as yet unrecognized driver mutations. They use a previously developed method to estimate mutation hotspots across the genome (Dig, Sherman et al 2022). This publication has already used cancer sequence data to infer driver mutations based on higher-than-expected mutation frequencies. The advance here is to further illustrate that recurrent mutations (estimated at 3 or more mutations (CDNs) at the same base) are more likely to be the result of selection for a driver mutation (Figure 3). Further analysis indicates that mutation sequence context (Figure 4) or mutation mechanisms (Figure 5) are unlikely to be major causes for recurrent point mutations. Finally, they calculate (Figure 6) that most driver mutations identifiable by the CDN approach could be identified with about 100,000 to one million tumor coding genomes.Weaknesses:The manuscript does provide specific examples where recurrent mutations identify known driver mutations but do not identify "new" candidate driver mutations. Driver mutation validation is difficult and at least clinically, frequency (ie observed in multiple other cancer samples) is indeed commonly used to judge if an SNV has driver potential. The method would miss alternative ways to trigger driver alterations (translocations, indels, epigenetic, CNVs). Nevertheless, the value of the manuscript is its quantitative analysis of why mutation frequencies can identify cancer driver mutations.Recommendations For The AuthorsWhereas the analysis of driver mutations in WES has been extensive, the application of the method to WGS data (ie the noncoding regions) would provide new information.

We appreciate that Reviewer #2 has suggested the potential application of our method to noncoding regions. Currently, the background mutation model is based on the site level mutations in coding regions, which hinders its direct applications in other mutation types such as CNVs, translocations and indels. We acknowledge that the proportion of patients with driver event involving CNV (73%) is comparable to that of coding point mutations (76%) as reported in the PCAWG analysis (Fig. 2A from Campbell et al., 2020). In future studies, we will attempt to establish a CNV-based background mutation rate model to identify positive selection signals driving tumorigenesis.

References

Abascal F, Harvey LMR, Mitchell E, Lawson ARJ, Lensing SV, Ellis P, Russell AJC, Alcantara RE, Baez-Ortega A, Wang Y, et al. 2021. Somatic mutation landscapes at single-molecule resolution. *Nature*:1–6.

Alexandrov LB, Kim J, Haradhvala NJ, Huang MN, Tian Ng AW, Wu Y, Boot A, Covington KR, Gordenin DA, Bergstrom EN, et al. 2020. The repertoire of mutational signatures in human cancer. *Nature* 578:94–101.

Bailey MH, Tokheim C, Porta-Pardo E, Sengupta S, Bertrand D, Weerasinghe A, Colaprico A, Wendl MC, Kim J, Reardon B, et al. 2018. Comprehensive Characterization of Cancer Driver Genes and Mutations. *Cell* 173:371-385.e18.

Baylin SB, Jones PA. 2016. Epigenetic Determinants of Cancer. *Cold Spring Harb Perspect Biol* 8:a019505.

Campbell PJ, Getz G, Korbel JO, Stuart JM, Jennings JL, Stein LD, Perry MD, Nahal-Bose HK, Ouellette BFF, Li CH, et al. 2020. Pan-cancer analysis of whole genomes. *Nature* 578:82–93.

Drost J, Clevers H. 2018. Organoids in cancer research. *Nat Rev Cancer* 18:407–418.

Elliott K, Larsson E. 2021. Non-coding driver mutations in human cancer. *Nat Rev Cancer* 21:500–509.

Grzeskowiak CL, Kundu ST, Mo X, Ivanov AA, Zagorodna O, Lu H, Chapple RH, Tsang YH, Moreno D, Mosqueda M, et al. 2018. In vivo screening identifies GATAD2B as a metastasis driver in KRAS-driven lung cancer. *Nat Commun* 9:2732.

Hess JM, Bernards A, Kim J, Miller M, Taylor-Weiner A, Haradhvala NJ, Lawrence MS, Getz G. 2019. Passenger Hotspot Mutations in Cancer. *Cancer Cell* 36:288-301.e14.

Hodis E, Triglia ET, Kwon JYH, Biancalani T, Zakka LR, Parkar S, Hütter J-C, Buffoni L, Delorey TM, Phillips D, et al. 2022. Stepwise-edited, human melanoma models reveal mutations’ effect on tumor and microenvironment. *Science* 376:eabi8175.

Lawrence MS, Stojanov P, Polak P, Kryukov GV, Cibulskis K, Sivachenko A, Carter SL, Stewart C, Mermel CH, Roberts SA, et al. 2013. Mutational heterogeneity in cancer and the search for new cancer-associated genes. *Nature* 499:214–218.

Makova KD, Hardison RC. 2015. The effects of chromatin organization on variation in mutation rates in the genome. *Nat Rev Genet* 16:213–223.

Michels BE, Mosa MH, Streibl BI, Zhan T, Menche C, Abou-El-Ardat K, Darvishi T, Członka E, Wagner S, Winter J, et al. 2020. Pooled In Vitro and In Vivo CRISPR-Cas9 Screening Identifies Tumor Suppressors in Human Colon Organoids. *Cell Stem Cell* 26:782-792.e7.

Ng PK-S, Li J, Jeong KJ, Shao S, Chen H, Tsang YH, Sengupta S, Wang Z, Bhavana VH, Tran R, et al. 2018. Systematic Functional Annotation of Somatic Mutations in Cancer. *Cancer Cell* 33:450-462.e10.

Sherman MA, Yaari AU, Priebe O, Dietlein F, Loh P-R, Berger B. 2022. Genome-wide mapping of somatic mutation rates uncovers drivers of cancer. *Nat Biotechnol* 40:1634–1643.

Stamatoyannopoulos JA, Adzhubei I, Thurman RE, Kryukov GV, Mirkin SM, Sunyaev SR. 2009. Human mutation rate associated with DNA replication timing. *Nat Genet* 41:393–395.